# All-atom molecular dynamics simulations of Synaptotagmin-SNARE-complexin complexes bridging a vesicle and a flat lipid bilayer

Josep Rizo[1,2,3]*, Levent Sari[1,4], Yife Qi[5], Wonpil Im[6,7,8,9], Milo M Lin[1,4]

[1]Department of Biophysics, The University of Texas Southwestern Medical Center, Dallas, United States; [2]Department of Biochemistry, University of Texas Southwestern Medical Center, Dallas, United States; [3]Department of Pharmacology, University of Texas Southwestern Medical Center, Dallas, United States; [4]Green Center for Systems Biology, The University of Texas Southwestern Medical Center, Dallas, United States; [5]Department of Medicinal Chemistry, School of Pharmacy, Fudan University, Shanghai, China; [6]Department of Biological Sciences, Lehigh University, Bethlehem, United States; [7]Department of Chemistry, Lehigh University, Bethlehem, United States; [8]Department of Bioengineering, Lehigh University, Bethlehem, United States; [9]Department of Computer Science and Engineering, Lehigh University, Bethlehem, United States

*For correspondence: Jose.Rizo-Rey@UTSouthwestern.edu

**Abstract** Synaptic vesicles are primed into a state that is ready for fast neurotransmitter release upon $Ca^{2+}$-binding to Synaptotagmin-1. This state likely includes trans-SNARE complexes between the vesicle and plasma membranes that are bound to Synaptotagmin-1 and complexins. However, the nature of this state and the steps leading to membrane fusion are unclear, in part because of the difficulty of studying this dynamic process experimentally. To shed light into these questions, we performed all-atom molecular dynamics simulations of systems containing trans-SNARE complexes between two flat bilayers or a vesicle and a flat bilayer with or without fragments of Synaptotagmin-1 and/or complexin-1. Our results need to be interpreted with caution because of the limited simulation times and the absence of key components, but suggest mechanistic features that may control release and help visualize potential states of the primed Synaptotagmin-1-SNARE-complexin-1 complex. The simulations suggest that SNAREs alone induce formation of extended membrane-membrane contact interfaces that may fuse slowly, and that the primed state contains macromolecular assemblies of trans-SNARE complexes bound to the Synaptotagmin-1 $C_2B$ domain and complexin-1 in a spring-loaded configuration that prevents premature membrane merger and formation of extended interfaces, but keeps the system ready for fast fusion upon $Ca^{2+}$ influx.

## Editor's evaluation

Using all-atom molecular dynamics simulations to visualize the pre-fusion primed state during synaptic vesicle fusion is very original and this approach will certainly be used by others in the future. This work provides new insights into the protein organization prior to vesicle fusion that will help better understand the mechanisms of vesicle priming and evoked-release.

## Introduction

The release of neurotransmitters by $Ca^{2+}$-triggered synaptic vesicle exocytosis is key for communication between neurons. The high speed of this process arises in part because synaptic vesicles are first tethered at the plasma membrane and undergo priming processes that leave them ready for fast fusion when an action potential induces $Ca^{2+}$ influx (*Südhof, 2013*). Research for over three decades has led to extensive characterization of the neurotransmitter release machinery (*Brunger et al., 2018*; *Rizo, 2022*), allowing reconstitution of basic features of synaptic vesicle fusion with the central components of this machinery (*Lai et al., 2017*; *Liu et al., 2016*; *Ma et al., 2013*) and uncovering key aspects of the underlying mechanisms. The SNARE proteins syntaxin-1, SNAP-25 and synaptobrevin form a tight four-helix bundle called the SNARE complex that assembles (zippers) from the N- to the C-terminus to bring the vesicle and plasma membranes together, and is key for membrane fusion (*Hanson et al., 1997b*; *Poirier et al., 1998*; *Söllner et al., 1993*; *Sutton et al., 1998*). This complex is disassembled by NSF and SNAPs (no relation to SNAP-25) to recycle the SNAREs for another round of fusion (*Mayer et al., 1996*; *Söllner et al., 1993*). Munc18-1 and Munc13-1 play central roles in synaptic vesicle priming by organizing assembly of the SNARE complex via an NSF-SNAP-resistant pathway (*Ma et al., 2013*; *Prinslow et al., 2019*) whereby Munc18-1 first binds to a self-inhibited 'closed' conformation of syntaxin-1 (*Dulubova et al., 1999*; *Misura et al., 2000*). Munc13-1 later opens syntaxin-1 (*Ma et al., 2011*; *Yang et al., 2015*) while bridging the vesicle and plasma membranes (*Quade et al., 2019*; *Xu et al., 2017*), and Munc18-1 forms a template for SNARE complex assembly (*Baker et al., 2015*; *Jiao et al., 2018*; *Parisotto et al., 2014*; *Sitarska et al., 2017*).

Synaptotagmin-1 (Syt1) acts as the $Ca^{2+}$ sensor that triggers fast release (*Fernández-Chacón et al., 2001*) in a tight interplay with complexins whereby both Syt1 and complexins play inhibitory and active roles (*Giraudo et al., 2006*; *Reim et al., 2001*; *Schaub et al., 2006*; *Tang et al., 2006*). The $Ca^{2+}$ sensor function is performed by the two $C_2$ domains of Syt1 ($C_2$A and $C_2$B), which form most of its cytoplasmic region and bind multiple $Ca^{2+}$ ions via loops at the tip of β-sandwich structures (*Fernandez et al., 2001*; *Sutton et al., 1995*; *Ubach et al., 1998*; *Figure 1—figure supplement 1A*). These loops also mediate $Ca^{2+}$-dependent binding of both $C_2$ domains to phospholipids, which is critical for release (*Fernández-Chacón et al., 2001*; *Rhee et al., 2005*). The $C_2$B domain also binds to $PIP_2$ through a polybasic region on the side of the β-sandwich (*Bai et al., 2003*), which induces binding to the plasma membrane. Moreover, the $C_2$B domain can bind to the SNARE complex through three different surfaces (*Brewer et al., 2015*; *Zhou et al., 2015*; *Zhou et al., 2017*), although only binding through a so-called primary interface (*Zhou et al., 2015*) is firmly established as physiologically relevant (*Guan et al., 2017*; *Voleti et al., 2020*). Such binding is disrupted upon $Ca^{2+}$-dependent binding of Syt1 to $PIP_2$-containing membranes (*Voleti et al., 2020*).

Complexin-1 binds tightly to the SNARE complex through a central α-helix that is preceded by an accessory helix (*Chen et al., 2002*) and may play a stimulatory role in release by promoting formation of a primed state with enhanced release probability (*Chen et al., 2002*), by protecting trans-SNARE complexes from disassembly by NSF and αSNAP (*Prinslow et al., 2019*) and/or by synchronizing $Ca^{2+}$-triggered fusion mediated by the SNAREs and Syt1 (*Diao et al., 2012*). The complexin-1 accessory helix inhibits release (*Xue et al., 2007*), possibly because it causes steric clashes with the vesicle, hindering C-terminal assembly of the SNARE complex (*Radoff et al., 2014*; *Trimbuch et al., 2014*). These and other findings suggested that complexin-1 and Syt1 bind simultaneously to the SNARE complex in the primed state of synaptic vesicles, stabilizing this state and preventing premature fusion; in this model, $Ca^{2+}$ influx relieves the inhibition by inducing dissociation of Syt1 from the SNAREs and enabling cooperation between Syt1 and the SNAREs in promoting fusion (*Voleti et al., 2020*).

Despite this wealth of knowledge, fundamental questions remain about the mechanism of neurotransmitter release, particularly regarding how the SNAREs and Syt1 trigger fast, $Ca^{2+}$-dependent membrane fusion. Major hurdles to address this question are the dynamic nature of this process and the fact that the protein complexes that trigger fusion are assembled between two membranes. Although important clues on the nature of the primed macromolecular assembly have been obtained with structural studies of soluble proteins or complexes anchored on one membrane (*Chen et al., 2002*; *Grushin et al., 2019*; *Voleti et al., 2020*; *Zhou et al., 2015*), this assembly is most likely affected by its location between two membranes. This feature strongly hinders the possibility of crystallization, while application of NMR spectroscopy for structure elucidation is hampered by the large size of any reconstituted two-membrane system (*Voleti et al., 2021*). Conversely, the small size

of the SNAREs, Syt1 and complexin-1 hinders visualization by cryo-EM. Moreover, it is extremely challenging to capture transient states formed during the pathway to $Ca^{2+}$-triggered membrane fusion experimentally.

Molecular dynamics (MD) simulations offer a powerful tool to analyze dynamic biomolecular processes and model cellular membranes (*Marrink et al., 2019*). Simulations using continuum and/or coarse-grained representations have provided important insights into SNARE-mediated membrane fusion (*Fortoul et al., 2015*; *Kasson et al., 2006*; *Manca et al., 2019*; *McDargh et al., 2018*; *Mostafavi et al., 2017*; *Risselada et al., 2011*; *Sharma and Lindau, 2018*). Continuum models can access the longest timescales, but require experimental data or atomistic simulations to parameterize the material properties, and often need to constrain geometries or material properties due to lack of context-dependent parameters (*Fortoul et al., 2015*). Coarse-grained molecular simulation approaches are freed from some of these constraints but at the expense of reduced simulation speed, and are limited in their ability to capture certain entropic effects and protein conformational changes (see below). To date, coarse-grained models of SNARE-mediated fusion have accessed the low microsecond timescale (*Kasson et al., 2006*; *Risselada et al., 2011*; *Sharma and Lindau, 2018*). All-atom simulations are better suited to reproduce the finely-balanced network of interactions between proteins, $Ca^{2+}$, and lipids that are expected to lead to membrane fusion but, because of the large size of the systems involved (millions of atoms), the low microsecond time scale has only recently become accessible through the most powerful available high-performance computing resources. In this context, it is worth noting that the delay from $Ca^{2+}$ influx into the presynaptic terminal to observation of post-synaptic currents in rat cerebellar synapses at 38 °C is 60 μs (*Sabatini and Regehr, 1996*), and that multiple events occur within this time frame, including $Ca^{2+}$ binding to the sensor, release of inhibitory interactions that hinder premature fusion, $Ca^{2+}$-evoked synaptic vesicle fusion, opening of the fusion pore, diffusion of neurotransmitters through the synaptic cleft, binding of the neurotransmitters to their postsynaptic receptors and opening of the channels that underlie the postsynaptic currents. These observations suggest that the fusion step may occur in just a few microseconds and hence that it may be possible to recapitulate the initiation of $Ca^{2+}$-dependent synaptic vesicle fusions in all-atom MD simulations starting with a properly designed initial configuration.

Here, we present all-atom MD simulations with explicit water molecules of systems containing four trans-SNARE complexes bridging two flat bilayers or a vesicle and a flat bilayer, without or with fragments of Syt1 and/or complexin-1. Because of the limited simulation times and the absence of key components, our results cannot lead to definitive conclusions but they help visualize potential trajectories and intermediates along the pathway to fusion and reveal intriguing features, leading to predictions or hypotheses that can be tested experimentally and with additional simulations. Our data indicate that trans-SNARE complexes strongly pull two membranes together, as expected, but have a tendency to induce extended membrane-membrane adhesion interfaces that have been observed experimentally but fuse slowly (*Hernandez et al., 2012*; *Witkowska et al., 2021*). Our results also suggest that, in the primed state of synaptic vesicles, Syt1 and complexin-1 form a spring-loaded macromolecular assembly with trans-SNARE complexes that hinders formation of such extended contact interfactes and premature bilayer merger, but is ready for fast membrane fusion upon $Ca^{2+}$ influx.

## Results

### Four trans-SNARE complexes between two flat lipid bilayers

The possibility of observing membrane fusion in the low microsecond time scale in all-atom MD simulations depends critically on the choice of the starting configuration, but the exact nature of the primed state of synaptic vesicles is unknown. Hence, we used the structural and functional information available on this system to generate potential starting configurations. The MD simulations presented here involved systems ranging from 1.7 to 5.9 million atoms. While multiple replicas of each simulation should ideally be carried out to verify the consistency of the results, performing replicated simulations would have limited the number of systems that we could study. Moreover, each simulation included four-trans SNARE complexes bridging two lipid bilayers and the variability in the behavior of the complexes in each simulation already provided insights into the consistency of the observed behaviors. Hence, we chose to use the available high performance computing time to investigate

systems with different components and/or distinct geometry, designing each new starting configuration according to what we had learned from the previous simulations. The simulations generated a large amount of data and it is impossible to describe a thorough analysis within the constraints of a single paper. Here, we present the main observations from the analyses that we have performed, and key files from the simulations are available in Dryad for further analyses.

In all the systems that we built, the composition of the bilayer containing anchored synaptobrevin approximated the lipid composition of synaptic vesicles (*Takamori et al., 2006*) and that of the bilayer with anchored syntaxin-1 was based on the lipid composition of the plasma membrane (*Chan et al., 2012*; *Table 1*). Both bilayers had asymmetric lipid distributions in the two leaflets to mimic those present in vivo (*Kobayashi and Menon, 2018*). The first system that we built was designed to examine whether SNARE complexes alone can bend two flat lipid bilayers and initiate bilayer fusion. The system contained four trans-SNARE complexes between two square lipid bilayers. The number of SNARE complexes was based on symmetry considerations and the finding that fast vesicle fusion typically observed in synapses requires at least three SNARE complexes (*Mohrmann et al., 2010*). For simplicity, the SNARE complexes contained the four SNARE motifs, the transmembrane (TM) sequences of syntaxin-1 and synaptobrevin, and the juxtamembrane linkers between their respective SNARE motif and TM region, but did not include the syntaxin-1 N-terminal region or the long linker between the two SNAP-25 SNARE motifs.

A key aspect in the design of realistic potential states of trans-SNARE complexes is the conformation of the juxtamembrane linkers of syntaxin-1 and synaptobrevin. Popular models of SNARE-mediated membrane fusion depicted continuous helices spanning the SNARE motifs, juxtamembrane linkers and TM regions for both synaptobrevin and syntaxin-1, envisioning that these helices can bend to accommodate the geometry of trans-SNARE complexes (*Hanson et al., 1997a*; *Sutton et al., 1998*; *Weber et al., 1998*). These models were supported by coarse-grained MD simulations that used the MARTINI force field and modeled the SNAREs in continuous helical conformations (*Risselada et al., 2011*). However, the intrinsic helical restraints enforced by the force field might bias the results and/or obscure the potential role of conformational changes in the dynamical coupling of the SNAREs to membrane fusion. Moreover, the bending of the helices required to form trans-SNARE complexes leads to unrealistic conformations that are expected to be unfavorable energetically because of their distorted geometry and are not commonly observed in protein structures. Thus, the helical restraints might have played a key role in membrane fusion in these simulations. Although continuous helices were observed in the crystal structure of a cis-SNARE complex that represents the configuration occurring after membrane fusion (*Stein et al., 2009*), the natural expectation is that the helical structure must break somewhere to accommodate the geometry of a trans-SNARE complex, most likely at the juxtamembrane linker. This expectation has been supported experimentally (*Kim et al., 2002*) and with all-atom MD simulations (*Bykhovskaia, 2021*). Moreover, helix continuity in the linkers is not required for neurotransmitter release (*Kesavan et al., 2007*; *Zhou et al., 2013*).

Thus, to generate trans-SNARE complexes for our simulations we started with the crystal structure of the cis-SNARE complex but we did not impose restraints on the conformation of the juxtamembrane linkers. Since the N-terminal half of the SNARE four-helix bundle is more stable than the C-terminal half (*Chen et al., 2002*; *Gao et al., 2012*) and is more distal from the membrane, we imposed position restraints for only the N-terminal half. In addition, we used position restraints to force the TM regions of synaptobrevin and syntaxin-1 to designed locations for insertion in their corresponding bilayers.

A short (1 ns) restrained MD simulation in water was sufficient for this purpose and led to unstructured conformations for the juxtamembrane linkers without substantially altering the four-helix bundle even at the C-terminal half, which was not restrained (*Figure 1—figure supplement 1B*). Four copies of the resulting trans-SNARE complex were generated by translations and rotations (*Figure 1—figure supplement 1C*), and were merged with the two bilayers separated by 5 nm to generate the initial configuration of this system (*Figure 1A*). We then carried out an unrestrained production simulation of this system for 750 ns at 310 K. As expected, the two membranes became almost circular to minimize tension and were gradually drawn together by the SNAREs, although the minimal distance between the bilayers reached a plateau (*Figure 1—figure supplement 1D, E*). The two bilayers were actually drawn to each other on one side first (at about 110 ns, *Figure 1B*) and later on the other side (*Figure 1C*), leading to close packing of the lipids against the SNARE four-helix bundles (*Figure 1D*).

**Table 1.** Size in atoms, length of productions MD simulations, temperature, speed of the simulations on Frontera at TACC and lipid composition of the flat bilayers and the vesicle of the different systems.

| qscff | 1699436 atoms | | 750 ns | 310 K | | 24 ns/day with 16 nodes | | | |
|---|---|---|---|---|---|---|---|---|---|
| *Upper bilayer* | CHL1 | POPC | POPS | SAPE | SDPE | SDPS | total | | |
| Upper leaflet | 540 | 468 | 0 | 72 | 120 | 0 | 1200 | | |
| % | 45 | 39 | 0 | 6 | 10 | 0 | | | |
| Lower leaflet | 540 | 146 | 84 | 120 | 240 | 96 | 1226 | | |
| % | 44 | 11.9 | 6.9 | 9.8 | 19.6 | 7.8 | | | |
| *Lower bilayer* | CHL1 | POPC | POPS | SAPI2D | SAPE | SDPE | SDPS | total | |
| Upper leaflet | 540 | 134 | 120 | 60 | 84 | 168 | 120 | 1226 | |
| % | 44 | 11 | 9.8 | 4.9 | 6.9 | 13.7 | 9.8 | | |
| Lower leaflet | 540 | 538 | 0 | 0 | 48 | 84 | 0 | 1200 | |
| % | 45 | 44 | 0 | 0 | 4 | 7 | 0 | | |
| sqscff | 1700475 atoms | | 270 ns | 310 K 24 ns/day with 16 nodes | | | | | |
| qscv | 3222393 atoms | | 520 ns | 310 K | | 23 ns/day with 32 nodes | | | |
| | | | 454 ns | 325 K | | | | | |
| *Vesicle* | CHL1 | POPC | SAGL | SAPE | SAPI2D | SDPE | SDPS | SOPS | total |
| Outer leaflet | 1258 | 296 | 0 | 534 | 0 | 282 | 210 | 199 | 2779 |
| % | 45.3 | 10.6 | 0 | 19.2 | 0 | 10.1 | 7.6 | 7.2 | |
| Inner leaflet | 814 | 668 | 0 | 183 | 0 | 99 | 1 | 1 | 1766 |
| % | 46.1 | 37.8 | 0 | 10.4 | 0 | 5.6 | 0.1 | 0.1 | |
| *Flat bilayer* | CHL1 | POPC | SAGL | SAPE | SAPI2D | SDPE | SDPS | SOPS | total |
| Upper leaflet | 540 | 96 | 12 | 84 | 60 | 156 | 120 | 120 | 1188 |
| % | 45.4 | 8.1 | 1 | 7.1 | 5.1 | 13.1 | 10.1 | 10.1 | |
| Lower leaflet | 540 | 516 | 12 | 36 | 0 | 84 | 0 | 0 | 1188 |
| % | 45.4 | 43.4 | 1 | 3 | 0 | 7.1 | 0 | 0 | |
| prsg | 5056443 atoms | | 336 ns | 310 K | | 13 ns/day with 32 nodes | | | |
| *Flat bilayer* | CHL1 | POPC | SAGL | SAPE | SAPI2D | SDPE | SDPS | SOPS | total |
| Upper leaflet | 830 | 151 | 18 | 132 | 93 | 262 | 182 | 181 | 1849 |
| % | 44.9 | 8.2 | 1 | 7.1 | 5 | 14.2 | 9.8 | 9.8 | |
| Lower leaflet | 810 | 774 | 18 | 72 | 0 | 126 | 0 | 0 | 1800 |
| % | 45 | 43 | 1 | 4 | 0 | 7 | 0 | 0 | |
| prs2 | 5870280 atoms | | 310 ns | 310 K | | 16 ns/day 48 nodes | | | |
| *Flat bilayer* | CHL1 | POPC | SAGL | SAPE | SAPI2D | SDPE | SDPS | SOPS | total |
| Upper leaflet | 1035 | 184 | 23 | 161 | 115 | 322 | 230 | 230 | 2300 |
| % | 45 | 8 | 1 | 7 | 5 | 14 | 10 | 10 | |
| Lower leaflet | 1009 | 964 | 22 | 88 | 0 | 154 | 0 | 0 | 2237 |
| % | 45.1 | 43.1 | 1 | 3.9 | 0 | 6.9 | 0 | 0 | |
| prsncpxca | 5870246 atoms | | 439 ns | 310 K | | 16 ns/day 48 nodes | | | |

Vesicle: same as qscv system.

Flat bilayer: same as prs2 system.

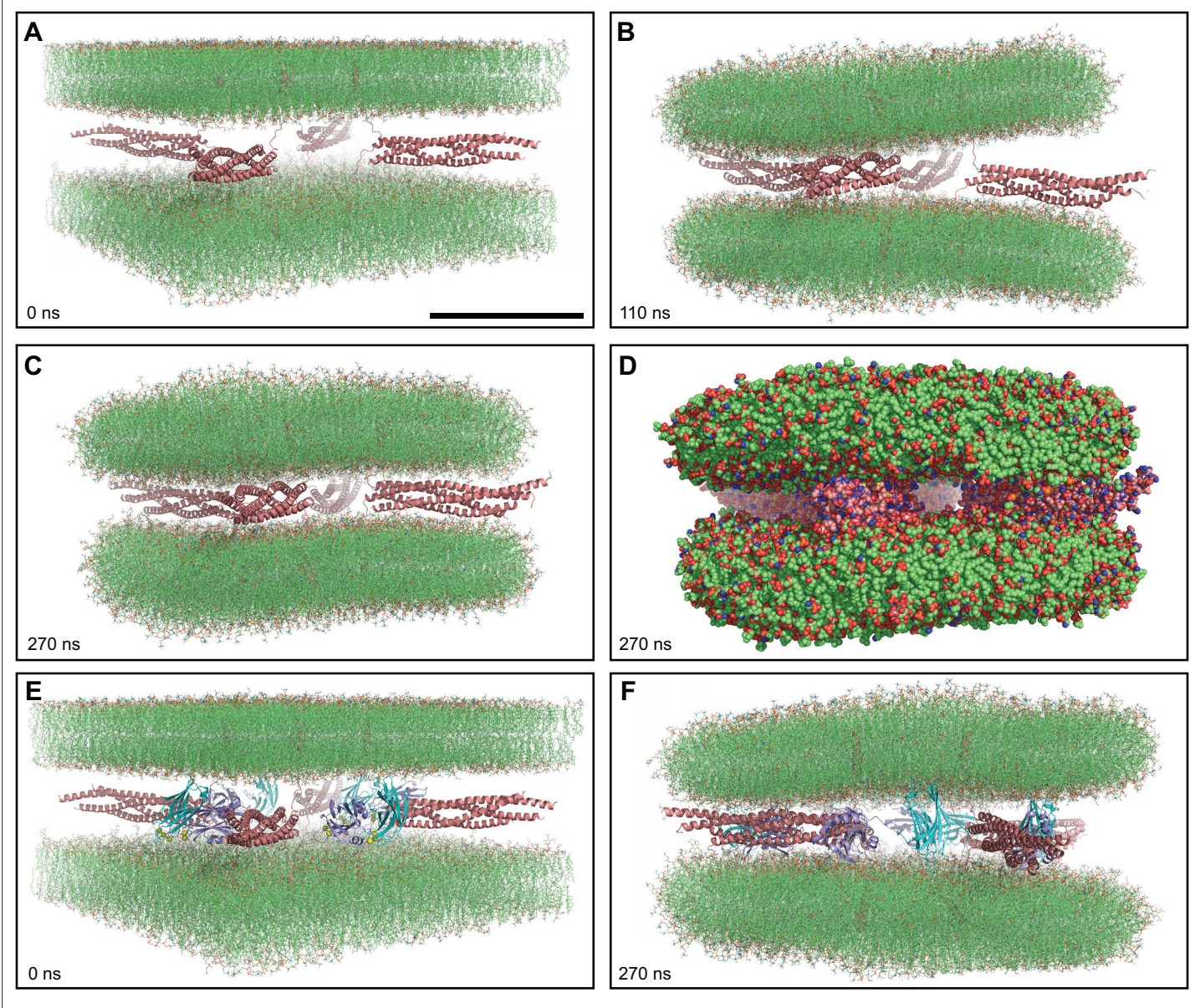

**Figure 1.** MD simulations of four trans-SNARE complexes bridging two flat bilayers. (A–C) Initial configuration of the system with SNARE complexes only (A), and snapshots of the MD simulation after 110 and 270 ns (B, C). The SNARE complexes are illustrated by ribbon diagrams in salmon. The lipids are shown as thin stick models. The scale bar in (A) equals 10 nm, which is a little shorter than the length of the SNARE four-helix bundle. (D) Snapshot of the same MD simulation at 270 ns showing all non-solvent atoms as spheres. (E–F) Initial configuration of the system containing four $Ca^{2+}$-bound Syt1 $C_2AB$ molecules in addition to the four trans-SNARE complexes (E) and snapshot of the simulation at 270 ns (F). SNARE complexes are illustrated by ribbon diagrams in salmon and the $C_2AB$ molecules are shown as ribbon diagrams with $C_2A$ in cyan and $C_2B$ in violet. The lipids are shown as thin stick models. The atom color code for the lipids is: carbon lime, oxygen red, nitrogen blue, phosphorous orange. $Ca^{2+}$ ions are shown as yellow spheres.

The online version of this article includes the following figure supplement(s) for figure 1:

**Figure supplement 1.** Set up of the system with four trans-SNARE complexes between two flat bilayers.

It is noteworthy that the four SNARE complexes were zippered at the C-terminus to the same extent as in the initial configuration and that extensive interactions were established between the juxtamembrane linkers and the membranes during the simulations. Such interactions were not unexpected, as both linkers contain abundant basic residues, the synaptobrevin linker in addition contains hydrophobic residues, and both linkers were shown to interact with the adjacent membrane (*Brewer et al., 2011*; *Kim et al., 2002*). Since much of the SNARE four-helix bundle is negatively charged, these

findings suggest that any electrostatic repulsion existing between the SNARE four-helix bundle and the membranes can be readily overcome by the high stability of the SNARE four-helix bundle and perhaps some contribution from the linker-bilayer interactions. During the 750 ns of the simulation we occasionally observed mild buckling of the syntaxin-1 membrane, but the buckling was reversible and there was no progress toward fusion. These findings suggest that four trans-SNARE complexes are unable to fuse two flat bilayers in the 1 µs time scale.

To explore whether Syt1 might cooperate with the SNAREs in bending two flat bilayers to initiate membrane fusion, we performed another simulation with an analogous system where we included a fragment spanning the two $C_2$ domains of Syt1 ($C_2AB$) bound to five $Ca^{2+}$ ions (*Fernandez et al., 2001*; *Ubach et al., 1998*; *Figure 1E*). During a 270 ns production MD simulation of this system, we observed that the $C_2AB$ molecules hindered the action of the trans-SNARE complexes in bringing the two bilayers closer (*Figure 1—figure supplement 1F*), particularly when the $C_2$ domains bind to one bilayer through the $Ca^{2+}$-binding loops and to the other bilayer via the opposite side of the β-sandwich, which is basic (*Figure 1F*). Although such bilayer-bilayer bridging might help in fusion (*Araç et al., 2006*) in a different configuration, it appeared that such potential action would require a much longer time scale in this system and we did not continue this simulation.

## Four trans-SNARE complexes bridging a vesicle and a flat bilayer

Based on the results from the simulations with four trans-SNARE complexes between two flat bilayers, we reasoned that, if synaptic vesicle fusion indeed occurs in the low microsecond time scale, this speed might require the geometry occurring at synapses, where small synaptic vesicles (ca. 40 nm diameter) fuse with the plasma membrane. To test this notion, we built a system with four trans-SNARE complexes (*Figure 2—figure supplement 1A*) bridging a vesicle and a flat bilayer (*Figure 2A*). The initial diameter of the vesicle (26 nm) was chosen as a compromise between making the system realistic and minimizing the overall size of the system to limit the time required for MD simulations. The vesicle was practically in molecular contact with the flat bilayer so that the system was poised for fusion. Since the lipid density of the vesicle was close to but not optimal, holes appeared in an initial production MD simulation. The holes were filled manually in an iterative process until the vesicle was stable (see Methods). During this procedure, the flat bilayer became circular and the vesicle became slightly smaller (24 nm diameter) (*Figure 2—figure supplement 1B, C*), but the diameter remained stable in subsequent production runs.

With the system equilibrated, we performed a production run of 520 ns at 310 K. Although we observed occasional flips of cholesterol molecules, there were no persistent perturbations of the bilayers that might signal the initiation of fusion. We raised the temperature to 325 K and carried out a production run of 454 ns in an attempt to accelerate fusion, but observed similar results. The final configuration illustrates that the vesicle diffused to some extent to one side with respect to the flat bilayer during the simulations (*Figure 2C*). The four-helix bundles of the four trans-SNARE complexes remained fully assembled up to the last hydrophobic layer [referred to as layer +8 (*Sutton et al., 1998*)] at the end of the simulations (*Figure 2D–G*), as in the initial configuration of the system (*Figure 2B*). One of the SNARE complexes became parallel to the flat bilayer (*Figure 2F*), whereas the other three had similar orientations as in the starting configuration (*Figure 2B, D, E and G*). The juxtamembrane linkers of synaptobrevin and syntaxin-1 established extensive interactions with the lipids early in the simulations. After 280 ns of the simulation at 310 K, we observed that the bottom of the vesicle was flattened, resulting in an extended contact interface with the flat bilayer (compare the slice view of *Figure 2J* with those of the initial configurations in *Figure 2H, I*). To corroborate these findings quantitatively, we calculated the number of contacts between oxygen atoms of the vesicle and the flat bilayer as a function of time. We assigned a contact to each oxygen-oxygen distance below 1 nm, which is a common cutoff used to calculate van der Waals and electrostatic interactions between atoms. The results showed that the number of contacts increased rapidly up to about 300 ns and then leveled off (*Figure 2—figure supplement 1D*). The extended contact interface persisted until the end of the simulation at 325 K, and during this simulation the flat bilayer became slightly curved to adapt to the shape of the vesicle (*Figure 2K*).

These finding correlates with results obtained in cryo-EM analyses of liposome fusion reactions mediated by the neuronal SNAREs, which revealed extended contact interfaces between the liposomes that are referred to as tight docking intermediates (*Hernandez et al., 2012*). These

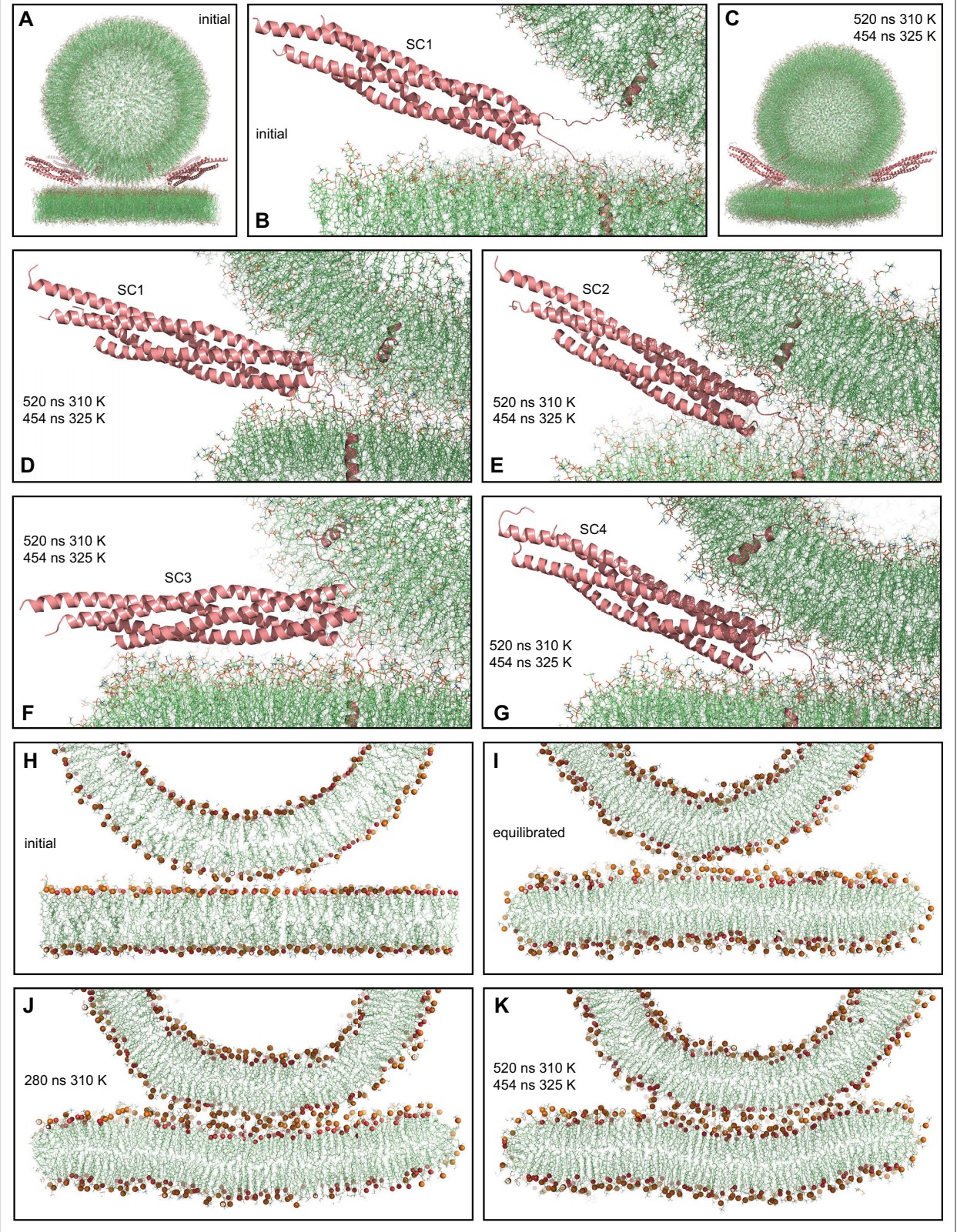

**Figure 2.** MD simulation of four trans-SNARE complexes bridging a vesicle and a flat bilayer. (**A**) Overall view of the initial system. (**B**) Close-up view of one of the trans-SNARE complexes in the initial system. (**C**) Snapshot of the system after a 520 ns MD simulation at 310 K and a 454 ns simulation at 325 K. (**D–G**) Close-up views of the four trans-SNARE complexes (named SC1-SC4) after the 520 ns MD simulation at 310 K and the 454 ns simulation at 325 K. In (**A–G**), the SNARE complexes are illustrated by ribbon diagrams in salmon. The lipids are shown as thin stick models (carbon lime, oxygen red,

*Figure 2 continued on next page*

*Figure 2 continued*

nitrogen blue, phosphorous orange). (**H–K**) Thin slices of the system in its initial configuration (**H**), after the equilibration steps (**I**), after 280 ns at 310 K (**J**) and after 520 ns at 310 K and 454 ns at 325 K (**K**). In (**H–K**) Phosphorous atoms of phospholipids and the oxygen atoms of cholesterol molecules are shown as spheres to illustrate the approximate locations of lipid head groups.

The online version of this article includes the following figure supplement(s) for figure 2:

**Figure supplement 1.** Set up of the system with four trans-SNARE complexes bridging a vesicle and a flat bilayer.

intermediates eventually evolve to yield membrane fusion, but fusion occurs in the second-minute time scale (*Hernandez et al., 2012*; *Witkowska et al., 2021*). Hence, it is unlikely that such extended interfaces occur in the pathway that leads to fast $Ca^{2+}$-triggered synaptic vesicle fusion. Note also that the energy required to initiate membrane fusion is expected to increase with the area of the interface between the two membranes, as larger areas require more lipid molecules to be rearranged. Interestingly, cryo-EM images of reconstitution reactions including additional components of the release machinery suggested that these additional components prevent formation of extended contact interfaces, favoring interfaces with smaller contact area between the bilayers that are referred to as point-of-contact interfaces (*Gipson et al., 2017*).

## Simulations of the primed Synaptotagmin-1-SNARE-complexin-1 complex

Overall, our simulations do not rule out the possibility that SNAREs alone might be able to induce membrane fusion in the low microsecond time scale, as it is plausible that other geometries might be more efficient in inducing fusion. However, the correlation of our results with the cryo-EM images of reconstitution experiments suggests that fast fusion requires additional proteins. Formation of a primed state of synaptic vesicles that is ready for fast release is the key to achieve fast $Ca^{2+}$-triggered fusion in synapses. The exact nature of this state is unclear, but Syt1 and complexin are most likely bound to trans-SNARE complexes in this state, as both proteins bind to the SNARE complex and are critical for fast, $Ca^{2+}$-triggered neurotransmitter release (*Fernández-Chacón et al., 2001*; *Reim et al., 2001*). A model of this state (*Voleti et al., 2020*) was proposed based on crystal structures of the SNARE complex bound to a complexin-1 fragment (*Chen et al., 2002*) or to the Syt1 $C_2B$ domain through the primary interface (*Zhou et al., 2015*), as well as a cryo-EM structure of Syt1 bound to lipid nanotube-anchored SNARE complex (*Grushin et al., 2019*). In this model, complexin-1 and the Syt 1 $C_2B$ domain bind to opposite sides of the SNARE four-helix bundle, and the $C_2B$ domain binds to the plasma membrane through a polybasic region on the side of the $C_2B$ domain opposite to the primary interface. However, the orientation of this macromolecular assembly with respect to the vesicle and plasma membranes, and the extent to which the SNARE complex is zippered, are unclear.

To gain insights into the nature of the primed Syt1-SNARE-complexin complex that is ready for fast $Ca^{2+}$-triggered membrane fusion, we built a system with four trans-SNARE complexes bridging a vesicle and a flat bilayer, each bound to a complexin-1 fragment and the Syt1 $C_2AB$ fragment as observed by crystallography (*Chen et al., 2002*; *Zhou et al., 2015*; below referred to as primed complexes). The complexin-1 fragment spanned residues 27–72 [Cpx1(27-72)], which include the central helix that binds to the SNARE complex and the preceding accessory helix that is believed to underlie the inhibitory activity of the accessory helix (*Trimbuch et al., 2014*; *Xue et al., 2007*). The system was designed to resemble a potential arrangement of the primed state, but implementing some flexibility such that the system could progress towards a preferred configuration of the proteins with respect to the two membranes. A restrained MD simulation performed to generate the initial protein arrangement led to partial unfolding of the C-terminal halves of the SNARE four-helix bundles but to distinct extents (*Figure 3A, B, D, F, H and J*, *Figure 3—figure supplement 1*), thus yielding a variety of starting configurations of the complexes.

After equilibration, we carried out a production simulation of 336 ns that resulted in the state shown in *Figure 3C*. Most substantial changes in the system occurred early in the simulation and each of the primed complexes appeared to reach a stable or metastable configuration by the end. The primed complexes exhibited some common behaviors and also distinct features. The SNARE four-helix bundle of one of the primed complexes (PC1) was almost fully assembled at the start of the simulation (up to layer +7, with a break in a SNAP-25 helix) and remained equally assembled at the end

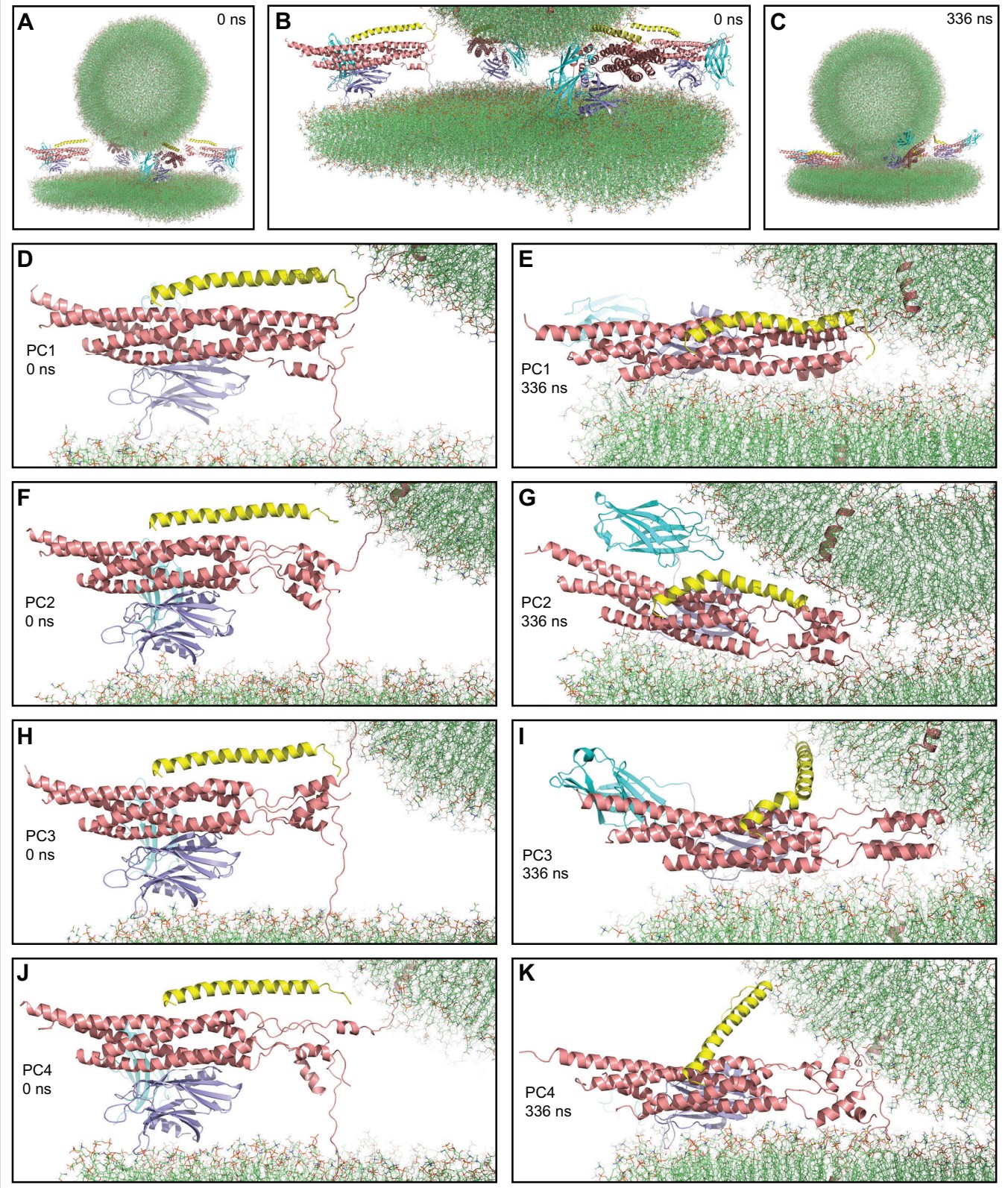

**Figure 3.** First MD simulation of primed complexes bridging a vesicle and a flat bilayer. (**A**) Overall view of the initial system after equilibration. (**B**) Close-up view of the four primed complexes in the initial system after equilibration. (**C**) Snapshot of the system after a 336 ns MD simulation. (**D–K**) Close-up views of the individual primed complexes (named PC1-PC4) in the initial configuration (**D,F,H,J**) and after the 336 ns MD simulation (**E,G,I,K**). In all panels, the primed complexes are illustrated by ribbon diagrams, with the SNAREs in salmon, Cpx1(27-72) in yellow and the Syt1 $C_2AB$

*Figure 3 continued on next page*

*Figure 3 continued*

fragment in cyan ($C_2A$ domain) and violet ($C_2B$ domain). The lipids are shown as thin stick models (carbon lime, oxygen red, nitrogen blue, phosphorous orange).

The online version of this article includes the following figure supplement(s) for figure 3:

**Figure supplement 1.** Ribbon diagrams of the four primed complexes generated for the first primed system with one vesicle and a flat bilayer.

**Figure supplement 2.** Additional views of the primed complexes bridging a vesicle and a flat bilayer.

(*Figure 3D, E*). The C-terminal halves of the SNARE four-helix bundles of the other three complexes were considerably more disrupted and, although they exhibited substantial changes during the simulation, they did not progress toward full assembly (*Figure 3F–K*). Interestingly, a few of the most C-terminal layers (+5 to+7) still formed a four-helix bundle in two complexes (PC2 and PC3, *Figure 3F-I*) and hence they may be particularly stable, but this feature did not seem to facilitate reassembly of the section of the four-helix bundle that was disrupted. Hence, although coil-to-helix transitions are known to occur very fast, in the 100 ns time scale (*Muñoz and Cerminara, 2016*), it appears that the constraints placed on the motions of the SNAREs in this complex system hinder the evolution toward a fully formed SNARE four-helix bundle.

Interestingly, the Cpx1(27-72) accessory helix exhibited clear steric clashes with the vesicle in all complexes. To avoid such clashes, the continuity between the central and accessory helices was broken in some cases, with the helix bending to one side or another (*Figure 3G, I*). In PC4, the entire helix changed orientation (*Figure 3K*), whereas in PC1, where the four-helix bundle is almost fully assembled, the helix was distorted into a snake shape (*Figure 3E*). The 'struggle' of the accessory helix to avoid bumps with the vesicle is particularly well illustrated by distinct bends of the Cpx1(27-72) helix occurring in PC3 during the simulations (*Figure 3—figure supplement 2A–E*). It is also noteworthy that Cpx1(27-72) remained bound to the SNAREs throughout the simulations due to interactions of the C-terminal end of the Cpx1(27-72) helix, particularly the Y70 aromatic ring, with a hydrophobic pocket of the SNARE complex, which persisted even when the overall direction of the helix changed in PC4 (*Figure 3—figure supplement 2F*). Overall, these observations provide a vivid visual illustration of the steric clashes between the complexin-1 accessory helix that may occur in the primed state, which were proposed to underlie the inhibition of neurotransmitter release caused by this helix (*Trimbuch et al., 2014*).

A common feature of the four primed complexes at the end of the simulation was the arrangement of the Syt1 $C_2B$ domain, which was initially placed between the SNARE four-helix bundle and the flat bilayer but in all primed complexes changed orientation, establishing extensive interactions between its polybasic face and the flat bilayer, and bringing the SNARE four-helix bundle close to the flat bilayer (*Figure 4A–D*). This arrangement dictates that the Cpx1(27-72) helix points toward the vesicle membrane, in agreement with the proposal that binding of Syt1 to the SNARE complex through the primary interface supports the inhibitory activity of Syt1 and complexin-1 (*Guan et al., 2017*; *Voleti et al., 2020*). The Syt1 $C_2A$ domain adopted distinct orientations in the different primed complexes, consistent with the fact that no stable $Ca^{2+}$-independent interactions of this domain with membranes or the SNARE complex have been identified. The $C_2B$ domain remained bound to the SNARE four-helix bundle via the primary interface in all four primed complexes throughout the simulation. The binding modes in the primed complexes resembled those observed in various crystal structures containing the primary interface (*Zhou et al., 2015*; *Zhou et al., 2017*), particularly in the so-called region I of this interface that includes Y338 among other side chains of $C_2B$ (e.g. *Figure 4E*). However, there were differences in the other region of this interface (region II), which includes R281, R398, and R399 of the $C_2B$ domain. In the crystal structures, there was variability in the contacts made by these side chains and R398 did not interact with acidic residues or was at moderate proximity with E238 of syntaxin-1 (*Figure 4—figure supplement 1A, B*). However, the R398 side chain interacted with a negative pocket formed by E55, D58 and E62 of SNAP-25 in the four primed complexes of our simulation (*Figure 4F*, *Figure 4—figure supplement 1C–E*). The findings that an R398Q mutation impairs binding of the $C_2B$ domain to the SNARE complex in vitro (*Voleti et al., 2020*) and disrupts neurotransmitter release in neurons (*Xue et al., 2008*) support the relevance of the interactions of R398 uncovered by our simulation and suggest that crystal packing might have slightly distorted the binding mode, but it is also plausible that the binding mode is dynamic in this area.

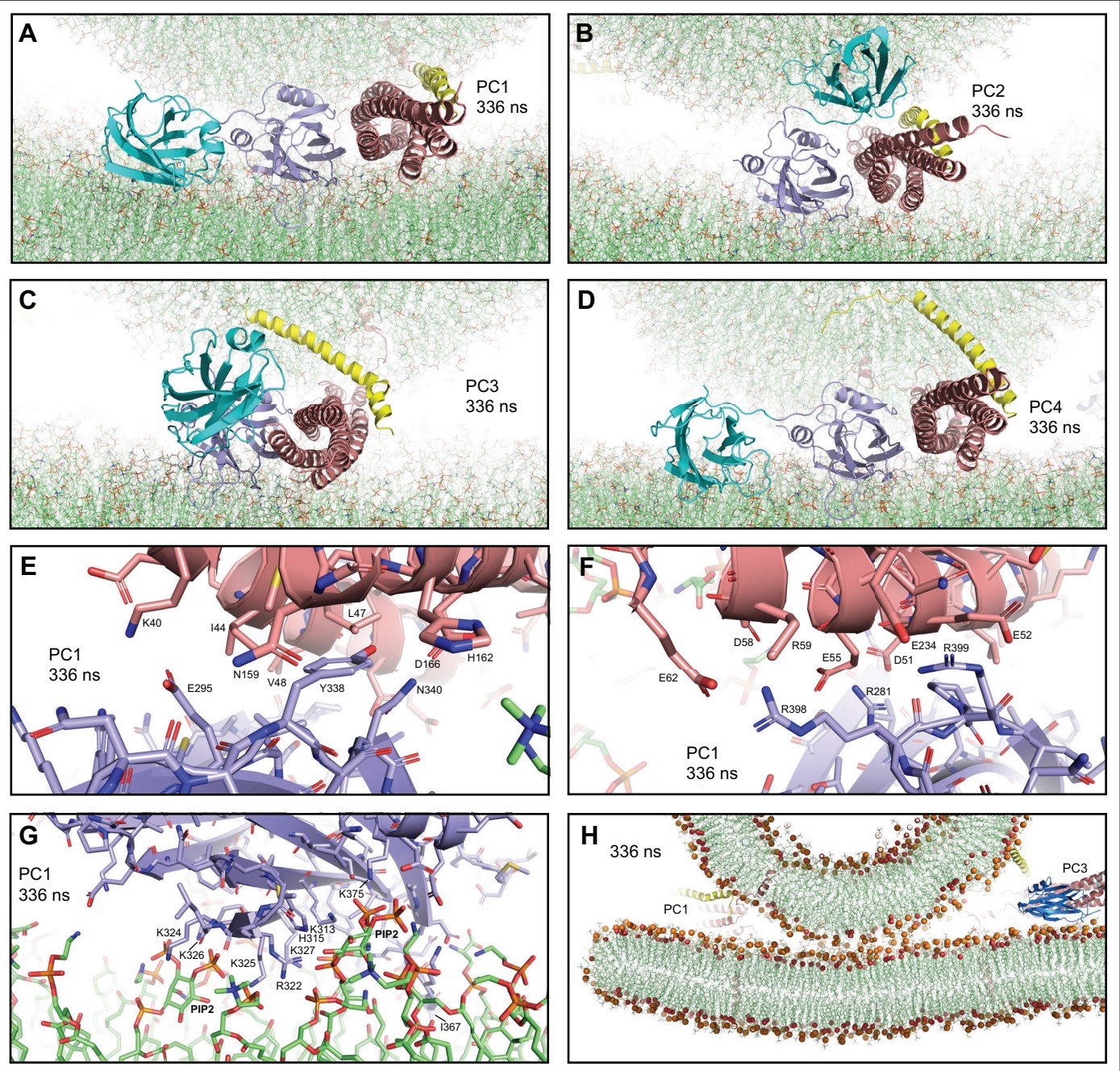

**Figure 4.** Additional views of the first MD simulation of primed complexes bridging a vesicle and a flat bilayer. (**A–D**) Close-up views of the four primed complexes after 336 ns showing how the Syt1 $C_2B$ domain binds to the SNARE complex through the primary interface and to the flat bilayer with the polybasic face, which dictates that the Cpx1(27-72) helix is oriented toward the vesicle and bends in different ways and directions to avoid steric clashes. This arrangement forces the SNARE four-helix bundle to be close to the flat bilayer. The primed complexes are illustrated by ribbon diagrams, with the SNAREs in salmon, Cpx1(27-72) in yellow and the Syt1 $C_2AB$ fragment in cyan ($C_2A$ domain) and violet ($C_2B$ domain). The lipids are shown as thin stick models (carbon lime, oxygen red, nitrogen blue, phosphorous orange). (**E–F**) Two different close-up views of the primary interface between the $C_2B$ domain and the SNARE complex in PC1 after 336 ns showing site I of the interface (**E**) or site II where R398,R399 of the $C_2B$ domain are located (**F**). The $C_2B$ domain and the SNARE complex are illustrated by ribbon diagrams and stick models with oxygen atoms in red, nitrogen atoms in blue, sulfur atoms in light orange and carbon atoms in violet [for the $C_2B$ domain] or salmon (for the SNAREs). The positions of selected side chains are indicated. (**G**) Close-up view of the interaction of the $C_2B$ domain of PC1 with the flat bilayer after 336 ns. The positions of $PIP_2$ headgroups, basic side chains involved in interactions with the lipids, and the hydrophobic side chain of I367 at the tip of a $Ca^{2+}$-binding loop that inserts into the bilayer, are indicated. (**H**) Thin slice of the system showing a point-of-contact interface between the vesicle and the flat bilayer at 336 ns. Phosphorous atoms of phospholipids

*Figure 4 continued on next page*

*Figure 4 continued*

and the oxygen atoms of cholesterol molecules are shown as spheres to illustrate the approximate locations of lipid head groups. The positions of PC1 and PC3 are indicated.

The online version of this article includes the following figure supplement(s) for figure 4:

**Figure supplement 1.** Additional close-up views of primed complexes bridging a vesicle and a flat bilayer.

**Figure supplement 2.** Second MD simulation of primed complexes bridging a vesicle and a flat bilayer.

**Figure supplement 3.** Additional views of the second MD simulation of primed complexes bridging a vesicle and a flat bilayer.

**Figure supplement 4.** Number of contacts in frames taken at 1 ns steps in the first (**A**) or second (**B**) simulation of four primed complexes bridging a vesicle and a flat bilayer.

In all four primed complexes, the extensive interactions of the $C_2B$ domain with the flat bilayer involved not only a polybasic sequence (residues 321–327) known to bind to $PIP_2$ (*Bai et al., 2003*) but also other basic residues on this face of the β-sandwich that are also important for neurotransmitter release [e.g. K313, (*Brewer et al., 2015*; *Figure 4G*, *Figure 4—figure supplement 1F-H*)]. $PIP_2$ molecules of the flat bilayer were often involved in these interactions. In addition, for all primed complexes, one of the $C_2B$ domain $Ca^{2+}$-binding loops interacted extensively with the flat bilayer, inserting the hydrophobic residue at its tip (I367) into the acyl region. We also observed some interactions of the flat bilayer with basic residues of the SNARE four-helix bundle (e.g. R30, R31 from synaptobrevin and R176 from SNAP-25), which appeared to be favored because the clashes between the Cpx1(27-72) helix and the vesicle push the SNARE four-helix bundle and $C_2AB$ toward the flat bilayer. There was some variability in the four-helix bundle-flat bilayer interactions observed in the different primed complexes, but the overall arrangement of the $C_2B$ domain with respect to the flat bilayer and the SNARE four-helix bundle was very similar in all complexes, regardless of the orientation of the Cpx1(27-72) helix and the state of assembly of the SNARE four-helix bundle at the C-terminus (*Figure 4A–D*).

To further test the consistency of our results with respect to the configuration of the primed Syt1-SNARE-complexin-1 complex, we built a similar system but using different configurations of the four initial primed complexes and a slightly larger square bilayer to provide more space for protein-membrane interactions. The final system (*Figure 4—figure supplement 2A, B*) was used to run a production simulation of 310 ns. *Figure 4—figure supplement 2C* shows the final configuration. The behaviors of the primed complexes were similar to those of the previous simulation. The four initial four-helix bundles again had different levels of assembly at the C-terminus, with PC1 being the only one that was almost completely assembled, and there was not much progress toward full assembly in the other three complexes (*Figure 4—figure supplement 2D-K*). The Cpx1(27-72) helix again exhibited strong clashes with the vesicle and distinct ways to overcome such clashes, whereas the $C_2B$ domain changed orientation to establish extensive interactions with the flat bilayer while remaining bound to the SNARE four-helix bundle via the primary interface (*Figure 4—figure supplement 2D*, *Figure 4—figure supplement 3A–D*). It is also noteworthy that, in the two simulations of primed complexes, contacts between the vesicle and the plasma membrane were established at about 210–230 ns and the contacts increased gradually afterwards, but appeared to be leveling off at the end of the simulations (*Figure 4—figure supplement 4*), resulting in point-of-contact interfaces between the vesicle and the flat bilayer, without flattening of the vesicle (*Figure 4H*, *Figure 4—figure supplement 3E*).

Overall, the arrangements of the Syt1 $C_2B$ domain with respect to the flat bilayer and the SNARE four-helix bundle in the eight primed complexes from the two simulations were very similar, and in all cases dictated that the Cpx1(27-72) helix was oriented toward the vesicle (*Figure 4A–D*, *Figure 4—figure supplement 3A–D*). The consistency of these results, together with the abundant data available on the functional importance of the $C_2B$-membrane, $C_2B$-SNARE and Cpx1(27-72)-SNARE interfaces present in these complexes [e.g. *Chen et al., 2002*; *Li et al., 2006*; *Zhou et al., 2015*] suggest that these complexes resemble those present in the primed state of synaptic vesicles.

## Simulation of the primed Synaptotagmin-1-SNARE-complexin-1 complex in the presence of Ca$^{2+}$

Ca$^{2+}$ binding to Syt1 is believed to induce a tight, PIP$_2$-dependent interaction of the C$_2$B domain with the plasma membrane and dissociation from the SNARE complex to relieve the inhibition of release caused by Syt1 and complexin-1 (*Voleti et al., 2020*). Based on the estimated $k_D$ of the interaction between the C$_2$B domain and the SNARE complex [ca. 20 μM *Voleti et al., 2020*], the off rate for dissociation is expected to be at most 2000 Hz and hence too slow for the time scales reachable in our simulations. However, it is plausible that dissociation might be strongly accelerated by changes in the orientation of the C$_2$B domain with respect to the membrane induced by Ca$^{2+}$ (*Voleti et al., 2020*). To examine whether we could observe the dissociation step and investigate how the system evolves afterwards through MD simulations, we generated a system analogous to that used for our first simulation of primed complexes, but with the larger flat bilayer used for the second simulation of primed complexes to provide sufficient room for Ca$^{2+}$-dependent binding to the C$_2$ domains. We added five Ca$^{2+}$ ions to the corresponding binding sites of C$_2$A and C$_2$B, and we removed the Cpx1(27-72) molecules to facilitate potential eventual fusion and to study at the same time how the system evolves without complexin-1 (*Figure 5A*).

We performed a production MD simulation of 439 ns, which led to the final configuration shown in *Figure 5B*. The C$_2$AB-SNARE complexes generally behaved similarly to the primed complexes in the previous simulations, but with some differences. The SNARE four-helix bundle that was almost fully assembled in the starting configuration remained assembled almost completely (up to layer +7), whereas the other three complexes did not make much progress toward C-terminal assembly (*Figure 5C–J*). We note again that 439 ns are expected to provide ample time for helix formation and large conformational rearrangements, which is exemplified by the behavior of one of the SNARE four-helix bundles (SC4) during the simulation. Thus, the helix corresponding to the SNAP-25 C-terminal SNARE motif was almost fully formed after 5 ns, even though there was a substantial break in the helix in the beginning, and there were considerable structural changes at 75 ns, but only limited changes from 75 to 439 ns (*Figure 5—figure supplement 1A–D*). These findings again show that the constraints imposed by the system hinder fast assembly of the C-terminus of the SNARE four-helix bundle. Interestingly, the SNARE four-helix bundles exhibited less interactions with the flat bilayer than in the simulations of primed complexes including Cpx1(27-72), consistent with the notion that the steric clashes of the complexin-1 accessory helix with the vesicle push the SNARE four-helix bundle toward the flat bilayer. As observed in previous simulations, the C$_2$B domains of the four complexes established extensive interactions with the flat bilayer and remained bound to the SNARE complex through the primary interface (*Figure 5D, F, H and J*). We did observe that SC4 became detached from R398,R399 of the C$_2$B domain early in the simulation and there were additional interactions in region I of the primary interface that remained at the end of the simulation (*Figure 5—figure supplement 1E, F*). However, it is unclear whether this change was caused by Ca$^{2+}$ binding to the C$_2$B domain. These findings suggest that dissociation of the C$_2$B domain from the SNAREs requires longer time scales and may be a rate limiting step in release, which is supported by the finding that an E295A/Y338W in the C$_2$B domain primary interface enhances SNARE complex binding (*Voleti et al., 2020*) but disrupts Ca$^{2+}$-evoked neurotransmitter release (*Zhou et al., 2015*). In this simulation, the vesicle came into contact with the flat bilayer at about 400 ns and the number contacts increased gradually afterwards but without reaching a plateau at the end (*Figure 5—figure supplement 2*).

## Discussion

Enormous advances have been made to elucidate the molecular mechanisms underlying neurotransmitter release and have suggested that Syt1 and complexin are bound to trans-SNARE complexes in the primed state that renders synaptic vesicles ready for fast fusion upon Ca$^{2+}$ influx. However, the configuration of the resulting macromolecular assembly is still unclear. Our MD simulations, together with previously available data, suggest that trans-SNARE complexes alone induce extended vesicle-plasma membrane contact interfaces that fuse slowly. Our results also indicate that binding of Syt1 and complexin-1 to trans-SNARE complexes in primed vesicles leads to a spring-loaded arrangement that hinders formation of such extended contact interfaces, keeping the system ready for fast fusion but at the same time hindering premature fusion before Ca$^{2+}$ influx.

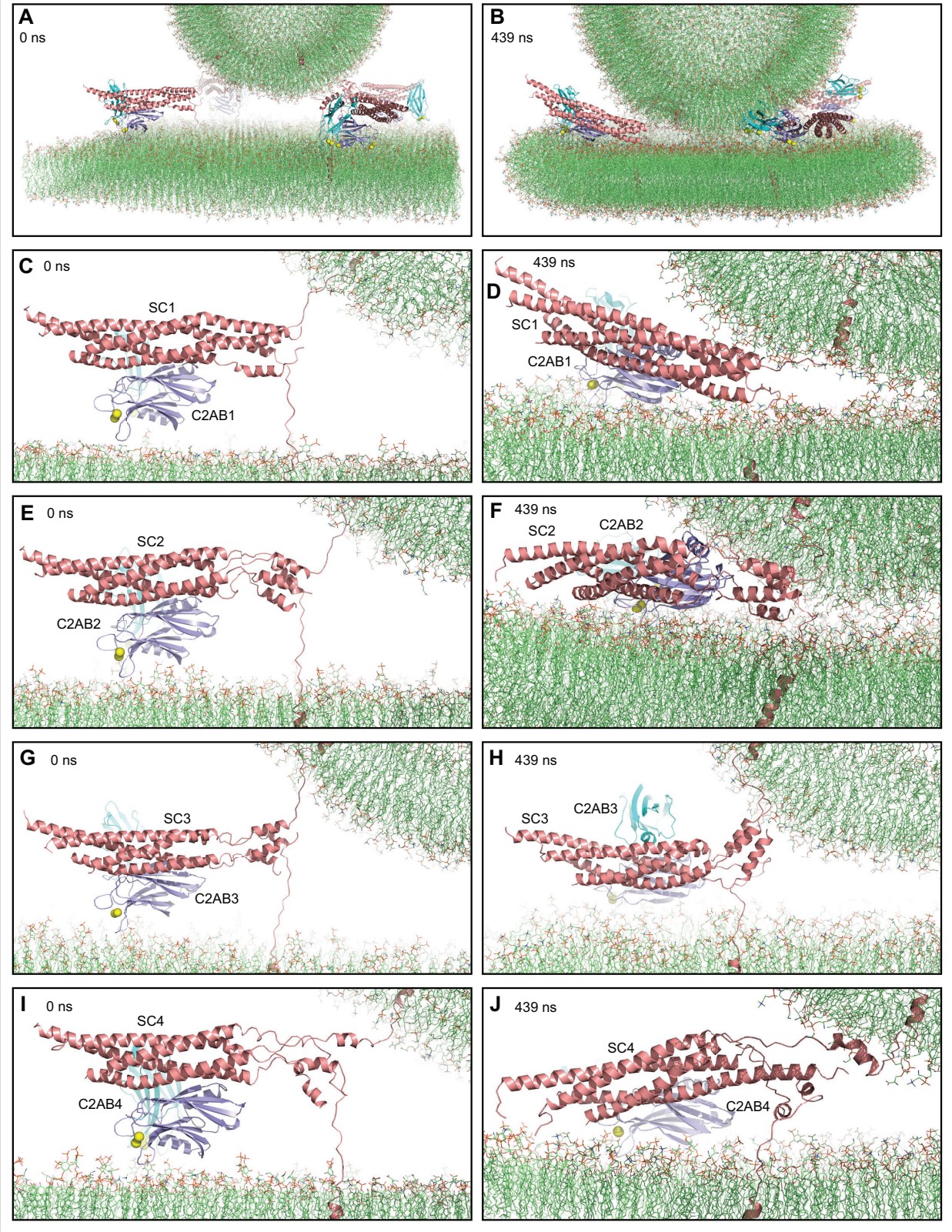

**Figure 5.** MD simulation of C$_2$AB bound to Ca$^{2+}$ and to trans-SNARE complexes bridging a vesicle and a flat bilayer. (**A**) Close-up view of the four C$_2$AB-SNARE complexes in the initial system. (**B**) Close-up view of the system after a 439 ns MD simulation. (**C–J**) Close-up views of the assemblies between C$_2$AB molecules (named C2AB1-4) and SNARE complexes (SC1-SC4) in the initial configuration (**C,E,G,I**) and after the 439 ns MD simulation (**D,F,H,J**). In all panels, the SNAREs are represented by ribbon diagrams in salmon and the Syt1 C$_2$AB fragment by ribbon diagrams in cyan (C$_2$A domain) and violet

*Figure 5 continued*

($C_2B$ domain). $Ca^{2+}$ ions are shown as yellow spheres. The lipids are shown as thin stick models (carbon lime, oxygen red, nitrogen blue, phosphorous orange).

The online version of this article includes the following figure supplement(s) for figure 5:

**Figure supplement 1.** Additional views of the complexes from *Figure 5*.

**Figure supplement 2.** Number of contacts in frames taken at 1 ns steps in the MD simulation of $C_2AB$ bound to $Ca^{2+}$ and to trans-SNARE complexes bridging a vesicle and a flat bilayer.

Our results need to be interpreted with caution because of the limited simulation times, the dependence of the results on the initial configurations and the absence of key elements of the release machinery. Nevertheless, multiple features observed in the simulations make sense from structural and energetic points of view, and are consistent with abundant experimental data available on this system. The SNARE four-helix bundle strongly drew the membranes together in our simulations with SNAREs alone (*Figure 1A–D*), in agreement with the high stability of the four-helix bundle [e.g. *Gao et al., 2012*]. However, assembly of the four-helix bundle brought the flat bilayers within a few nm from each other, and it is unclear how the SNAREs exert additional force on the membranes to induce fusion. A major problem with the widespread notion that the SNARE motif, juxtamembrane linker and TM region of synaptobrevin and syntaxin-1 form continuous helices that force fusion as they zipper from the N- to the C-terminus (*Hanson et al., 1997a*; *Sutton et al., 1998*; *Weber et al., 1998*) is that the bent conformations of the linkers envisioned in these models are unrealistic from an energetic point of view. Although optical tweezer data suggested that interactions between the juxtamembrane linkers contribute to exerting force on the membranes to induce fusion (*Gao et al., 2012*) helix continuity in the linkers is not required for neurotransmitter release (*Kesavan et al., 2007*; *Zhou et al., 2013*). Substantial release was observed even upon insertion of a five-residue sequence into the synaptobrevin linker (*Kesavan et al., 2007*) despite the fact that this sequence should break the register of linker-linker interactions and contained two (helix disrupting) glycine residues. Note also that the optical tweezer data were obtained in the absence of membranes and that, in vivo, the linkers are likely to interact with the lipids given the proximity of each linker to the adjacent membrane and the abundance of basic residues in the linker sequence (and aromatic residues in the case of synaptobrevin). The extensive interactions of the linkers with the membranes observed in all our simulations support this prediction.

Based on conformational grounds, it is not surprising that the juxtamembrane linkers became unstructured during the simulation that we performed to generate a trans-SNARE complex starting from the crystal structure of the cis-SNARE complex. Note however that we did not perform a systematic analysis to examine the range of linker structures that are compatible with the geometry of a trans-SNARE complex. In any case, the linkers are expected to be unstructured before SNARE complex assembly, which is supported by EPR data (*Kim et al., 2002*). Therefore, configurations with unstructured linkers were natural, unbiased starting points for the simulations, and helical conformation could be adopted by the linkers during the simulations if they were preferred. However, the linkers remained unstructured during all our simulations, which facilitated the extensive linker-membrane interactions observed. These interactions may have contributed to pull the two flat bilayers together in our first simulation (*Figure 1A–D*) and to induce the formation of an extended contact interface in the simulation with a vesicle and a flat bilayer (*Figure 2J*). Coarse-grained simulations have suggested that formation of such interfaces can also arise from entropic forces that favor outward movement of the SNARE complexes, away from the center of the interface (*Mostafavi et al., 2017*). These extended interfaces have been observed by cryo-EM and by fluorescence microscopy, and evolve to fusion in long time scales (seconds-minutes) (*Diao et al., 2012*; *Hernandez et al., 2012*; *Witkowska et al., 2021*), in agreement with data showing that liposome fusion occurs minutes after liposome docking (*Cypionka et al., 2009*). SNARE-mediated fusion was also observed at faster time scales (*Domanska et al., 2009*; *Heo et al., 2021*). Thus, it is plausible that fusion is slower under conditions that favor formation of extended interfaces. Interestingly, cryo-EM studies indicated that formation of such extended interfaces is hindered by other proteins involved in $Ca^{2+}$-evoked release, which favor point-of-contact interfaces that fuse faster (*Diao et al., 2012*; *Gipson et al., 2017*).

Our two simulations including Syt1 $C_2AB$ and the complexin-1 (27–72) fragment are consistent with this proposal, as these proteins appeared to hinder formation of extended interfaces (*Figure 4—figure supplement 4*), but longer simulations will be required to further test this notion. The initial configurations used in these simulations (*Figure 3A*, *Figure 4—figure supplement 2A*) were built to mimic and investigate potential states of the Syt1-SNARE-complexin-1 macromolecular assemblies that are likely central components of the primed state of synaptic vesicles. Overwhelming evidence supports the physiological relevance of the binding modes of the SNARE complex to complexin-1 and to the primary interface of the Syt1 $C_2B$ domain used to build these initial configurations (*Chen et al., 2002*; *Guan et al., 2017*; *Xue et al., 2007*; *Zhou et al., 2015*). These binding modes were largely preserved in the eight primed complexes during the two simulations, although there were slight rearrangements in the primary interface that allow closer interactions of the critical R398 side chain of the $C_2B$ domain (*Xue et al., 2008*) with acidic residues from the SNARE complex (*Figure 4F*).

In the initial configurations, we placed the $C_2B$ domain between the SNARE complex and the flat bilayer to facilitate interactions of a polybasic β-strand (residues 321–327) with $PIP_2$ that are believed to mediate binding of the $C_2B$ domain to the plasma membrane (*Bai et al., 2003*). Interestingly, each of the eight primed complexes was re-oriented during the two simulations to enable extensive interactions of the $C_2B$ domain with the flat bilayer that involve not only the polybasic β-strand but also other basic residues from the same side of the β-sandwich that are also known to play a key role in release [e.g. K313; *Brewer et al., 2015*; *Figure 4G*, *Figure 4—figure supplement 1F–H*, *Figure 4A-D*]. Thus, the resulting $C_2B$ domain-flat bilayer binding mode is consistent with the physiological importance of multiple residues of the polybasic face (*Brewer et al., 2015*). This binding mode resembles that observed in previous all-atom simulations of a $C_2AB$-SNARE-complexin-1 assembly on a flat bilayer (*Bykhovskaia, 2021*) and makes sense from a physicochemical point of view given the numerous electrostatic interactions involved. It is also worth noting that the extent of C-terminal assembly of the SNARE four-helix bundle was distinct in each of the eight complexes. Thus, the orientation of the $C_2B$ domain-SNARE-Cpx1(27-72) assembly with respect to the membranes consistently observed in the eight complexes can be accommodated by different extents of C-terminal assembly. All these observations suggest that the $C_2B$ domain binds to the plasma membrane in a similar mode in the primed state of synaptic vesicles and that some of the configurations of the complexes visited in the simulations resemble those present in the primed state.

Due to the $C_2B$-flat bilayer interactions, the SNARE four-helix bundle came near the flat bilayer, with the $C_2B$ domain on one side rather than between the bundle and the flat bilayer, and with the helix of Cpx1(27-72) oriented toward the vesicle (*Figure 4A–D*, *Figure 4—figure supplement Figure 4—figure supplements 2 and 3A–D*). This orientation dictates that formation of a continuous, straight helix by Cpx1(27-72) leads to steric clashes with the vesicle if the SNARE four-helix bundle is fully or close to fully zippered, consistent with the proposal that such steric clashes underlie the inhibitory role of the accessory helix and with physiological data supporting this notion (*Trimbuch et al., 2014*). These steric clashes most likely cause, at least to some extent, the distinct ways by which the Cpx1(27-72) helix was bent in the different complexes (*Figures 3D–K and 4A–D*, *Figure 4—figure supplement 3A-D*). These variations may also arise in part from random motions and are consistent with data suggesting that the accessory helix can adopt additional orientations that depend on the extent of C-terminal zippering (*Choi et al., 2016*; *Kümmel et al., 2011*). However, NMR data showed that the accessory helix sequence has a high propensity to form α-helical conformation that nucleates the central helix and that structures with a continuous, straight helix as observed by X-ray crystallography are substantially populated when there is no steric hindrance (*Chen et al., 2002*; *Pabst et al., 2000*; *Radoff et al., 2014*). Overall, these observations show that, even if there are natural motions in the complexin-1 helix, transient formation of a continuous straight helix results in steric clashes with the vesicle if the four-helix bundle zippers fully.

It is difficult to estimate the energy barrier imposed by this behavior on SNARE zippering, but it is known that the energy cost of breaking just one hydrogen bond to distort the complexin-1 helix can be about 1.8 $k_BT$ (*Nick Pace et al., 2014*), which could translate to 7.2 $k_BT$ by cooperative action of four primed complexes. Such an energy barrier would slow down SNARE zippering by a factor of 1340 (=e^7.2). For comparison, the enhancements of spontaneous neurotransmitter release observed upon deletion of complexin range from 3 to >20 (*Huntwork and Littleton, 2007*; *Martin et al., 2011*). Thus, even a small energy barrier caused by steric hindrance can account for these enhancements.

Note also that we observed occasional interactions of the Cpx1(27-72) accessory helix with C-terminal residues of the synaptobrevin and SNAP-25 SNARE motifs that are favored by proximity (e.g. *Figure 3G*, *Figure 3—figure supplement 2G*), and recent cross-linking experiments suggested that very weak interactions of the complexin accessory helix with synaptobrevin and SNAP-25 hinder C-terminal zippering and release (*Malsam et al., 2020*). It is unlikely that specific interactions underlie the complexin inhibitory function, as this function was retained when the accessory helix was replaced with an unrelated helical sequence, and helix propensity appears to be the key determinant for the inhibitory function (*Radoff et al., 2014*). However, very weak complexin-SNARE interactions that do not need to be specific may slow down C-terminal SNARE zippering and thus contribute also to inhibition of release by the complex accessory helix.

While all these observations support the proposal that the consistent overall configuration of the Syt1 $C_2B$ domain-SNARE-Cpx1(27-72)-flat bilayer assembly observed for the eight primed complexes in the two simulations resembles that present in primed vesicles, there are clear uncertainties with regard to the extent of C-terminal SNARE zippering and the vesicle-plasma membrane distance, which are closely related. In our two simulations, the vesicle was drawn into contact with the flat bilayer. However, other factors in addition to complexin are likely to hinder zippering in vivo, most notably Munc13-1 because in the absence of $Ca^{2+}$ it bridges membranes in approximately perpendicular orientations that keep the membranes apart (*Camacho et al., 2021*; *Grushin et al., 2022*; *Quade et al., 2019*). Analyses by high-pressure freezing electron tomography (ET) (*Imig et al., 2014*) and cryo-ET (*Radhakrishnan et al., 2021*) showed that docked vesicles exhibit a distribution of distances from the plasma membrane that range from 0 to several nm, and a density map built from a subset of subtomograms revealed a distance of 3.5 nm. However, a recent model that can explain a large amount of available presynaptic plasticity data invoked two primed states, one that involves partially

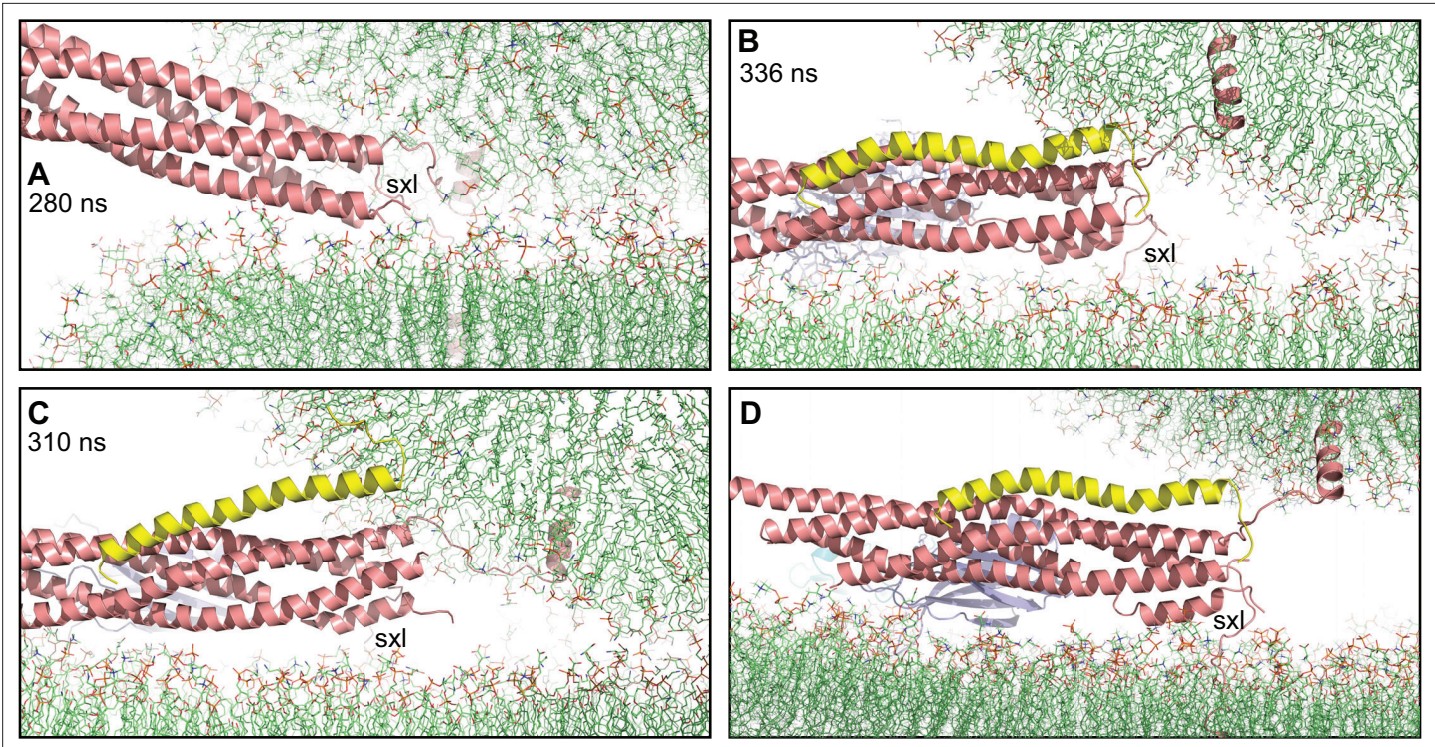

**Figure 6.** Complexin-1 may hinder the final action of trans-SNARE complexes to bring membranes together. (**A–C**) Close-up views of one of the SNARE complexes bridging a vesicle and a flat bilayer after simulation for 280 ns at 310 K (**A**) (shown in *Figure 2D* after 520 ns at 310 K and 454 ns at 325 K), of PC1 in the first MD simulation of primed complexes after 336 ns (**B**) (also shown in *Figure 3E*) and of PC1 in the second MD simulation of primed complexes after 310 ns (**C**) (also shown in *Figure 4—figure supplement 2E*). The complexes are illustrated by ribbon diagrams, with the SNAREs in salmon, Cpx1(27-72) in yellow and the Syt1 $C_2AB$ fragment in cyan ($C_2A$ domain) and violet ($C_2B$ domain). The positions of the syntaxin-1 juxtamembrane linkers (sxl) are indicated. The comparison shows how the SNARE complex with the fully assembled four-helix bundle in (**A**) drew the two membranes closer than the two primed complexes of (**B,C**). (**D**) Close up view of the pose shown in (**B**) after moving the vesicle upwards so that it's minimal distance from the flat bilayer is 3 nm.

assembled SNARE complexes and has low release probability (loose state), and another with more fully assembled SNARE complexes that has a much higher release probability (tight state; *Neher and Brose, 2018*). Hence, it is plausible that the vesicles that are closest to the plasma membrane account for much of the neurotransmitter release observed.

Based on these observations and our simulations, we envision two scenarios. In one scenario, vesicles in the tight primed state are very close to or in contact with the plasma membrane and include primed complexes that resemble PC1 at the end of each of our two simulations (*Figure 3E*, *Figure 4— figure supplement 2E*). In both of these poses, the four-helix bundle was almost fully zippered, but the vesicle was not as close to the flat bilayer around the C-terminus of the four-helix bundle as in the simulation with only SNAREs (*Figure 6A–C*), which led to an extended vesicle-flat bilayer interface (*Figure 2J*). Since complexin-1 binding stabilizes the C-terminus of the SNARE complex (*Chen et al., 2002*), it is tempting to speculate that, to maximize the speed of release, complexin-1 favors formation of an almost fully zippered, tight primed state that is ready for fast $Ca^{2+}$-triggered fusion, but hinders the 'final pull' of the SNARE four-helix bundle to bring the membranes together, preventing premature fusion as well as formation of an extended interface. In this spring-loaded model, dissociation of the $C_2B$ domain from the SNARE complex upon $Ca^{2+}$ influx (*Voleti et al., 2020*) allows rotations of the four-helix bundle that eliminate the steric clashes and facilitate cooperation between the Syt1 $C_2$ domains and the SNAREs in triggering fusion by a mechanism that remains unclear. Hence, this model can explain the finding that complexins are required for the dominant negative effect of mutations in the $Ca^{2+}$-binding site of the Syt1 $C_2B$ domain (*Zhou et al., 2017*). In the second scenario, the distance from the vesicle to the plasma membrane in the tight primed state is a few nm. Manually moving the vesicle in the pose of *Figure 6B* to a distance of 3 nm from the flat bilayer (*Figure 6D*) shows that the same spring-loaded configuration of the Syt1 $C_2B$ domain-SNARE-Cpx1(27-72)-flat bilayer assembly could be kept by stretching the synaptobrevin juxtamembrane linker, and the Cpx1(27-72) helix would still hinder progress toward final zippering and fusion. This configuration is also largely compatible with longer vesicle-flat bilayer distances if the linker is stretched further and/or there is partial SNARE unzipping. In this case, steric clashes of the Cpx1(27-72) helix with the vesicle might be alleviated or eliminated, but they would occur as soon there is full zippering and hence would still hinder vesicle fusion.

Clearly, these models will need to be tested with further MD simulations and experimentation, and there are multiple factors of the primed state of synaptic vesicles that were not included in the simulations. As mentioned above, particularly important for this state is Munc13-1, but other missing elements include Munc18-1, the N-terminal region of syntaxin-1, the SNAP-25 linker joining its SNARE motifs, the N- and C-terminal regions of complexin-1 and the TM region plus linker sequence of Syt1 [reviewed in *Rizo, 2022*]. Moreover, the Syt1 $C_2B$ domain can also bind to the SNARE complex through a so-called tripartite interface whereby an α-helix of the $C_2B$ domain is adjacent to the complexin-1 helix (*Zhou et al., 2017*), which could stabilize the complexin-1-SNARE interface to hinder rotations that might lead to dissociation (see *Figure 3K*, *Figure 3—figure supplement 2F*). There is also evidence that Syt1 forms oligomeric rings that hinder spontaneous neurotransmitter release (*Tagliatti et al., 2020*), and it is plausible that the SNAREs alone can induce fast fusion in different configurations that we did not study. Furthermore, some evidence suggested many years ago that direct binding of $Ca^{2+}$ to phospholipids could trigger synaptic vesicle fusion (*Papahadjopoulos et al., 1976*) and, although it is now generally believed that $Ca^{2+}$ triggers neurotransmitter release by binding to Syt1, it is plausible that $Ca^{2+}$-phospholipid interactions might also contribute to trigger membrane fusion. Addressing all these issues with all-atom MD simulations will be challenging because of the limited simulation times that are currently reachable. Continuum and coarse-grained simulations, which have already provided important insights into SNARE-mediated membrane fusion (*Fortoul et al., 2015*; *Manca et al., 2019*; *McDargh et al., 2018*; *Mostafavi et al., 2017*; *Risselada et al., 2011*; *Sharma and Lindau, 2018*), offer the opportunity to explore much longer time scales with similar systems. Hence, a marriage of approaches whereby the most interesting results obtained by such simulations are investigated in further detail by all-atom simulations will likely provide a powerful strategy to unravel the intricate mechanisms that govern fast $Ca^{2+}$-triggered membrane fusion. The systems that we present here provide a framework to pursue these studies and gradually incorporate additional elements of the neurotransmitter release machinery.

## Methods

### High-performance computing

Most high-performance computing, including all production MD simulations, were performed using Gromacs (*Pronk et al., 2013*; *Van Der Spoel et al., 2005*) with the CHARMM36 force field (*Huang et al., 2017*; *Klauda et al., 2010*; *Lee et al., 2019*; *Wu et al., 2014a*; *Wu et al., 2014b*) on Frontera at TACC. Some of the initial setup tests, solvation, ion addition, minimizations, and equilibration steps were performed using Gromacs with the CHARMM36 force field at the BioHPC supercomputing facility of UT Southwestern, or on Lonestar5 or Stampede2 at TACC. System visualization and manual manipulation were performed with Pymol (Schrödinger, LLC).

### System setup

All systems were built by manually combining coordinates of the protein components with coordinates of the membranes, solvating the system with explicit water molecules (TIP3P model) and adding potassium and chloride ions as needed to reach a concentration of 145 mM and make the system neutral. Flat lipid bilayers were built with the Membrane Builder module (*Jo et al., 2007*; *Jo et al., 2009*) in the CHARMM-GUI (*Jo et al., 2008*) website (https://charmm-gui.org/), providing the coordinates of the TM region of synaptobrevin or syntaxin-1 in their desired positions as input. The bilayers contained mixtures of cholesterol (CHL1), 16:0-18:1 phosphatidylcholine (POPC), 18:0-22:6 phosphatidyltethanolamine (SDPE), 18:0-22:4 phosphatidyltethanolamine (SAPE), 18:0-18:1 phosphatidylserine (SOPS), 18:0-22:6 phosphatidylserine (SDPS), 18:0-20:4 phosphatidylinositol 4,5-bisphosphate (SAPI2D) and/or 18:0-20:4 glycerol (SAGL). *Table 1* list the number of atoms and lipid compositions of the membranes; each entry corresponds to a system denoted by the abbreviations described below, which were used as roots for the filenames of the corresponding simulations.

*Four trans-SNARE complexes bridging two flat bilayers* (qscff system). The starting point to generate a trans-SNARE complex was the crystal structure of the neuronal SNARE complex that included the TM regions of synaptobrevin and syntaxin-1 (*Stein et al., 2009*). Two residues at the C-terminus of syntaxin-1 (residues 287–288) were added in Pymol, and four residues of the C-terminus of SNAP-25 (residues 201–204) were added manually based on a crystal structure of soluble SNARE complex (PDB accession code 1NS7). The resulting complex included residues 30–116 of synaptobrevin, residues 189–288 of syntaxin-1, and residues 8–82 and 141–204 of SNAP-25. To move the TM regions of synaptobrevin and syntaxin-1 to designed positions where they were later inserted into the flat lipid bilayers (*Figure 1—figure supplement 1B*), the cis-SNARE complex was solvated with explicit water molecules and energy minimized. Then a 1 ns production MD simulation at 310 K was performed imposing position restraints to keep of all heavy atoms of the N-terminal half of the SNARE complex, up to the polar layer (residues 30–56 of synaptobrevin, residues 189–226 of syntaxin-1, and residues 8–53 and 141–174 of SNAP-25), in their original coordinates (force constant 1,000 kJ/mol/nm$^2$), and to force the backbone atoms of the TM regions (residues 95–116 of synaptobrevin and residues 266–288 of syntaxin-1) to move to the designed positions (force constant 300 kJ/mol/nm$^2$). Four copies of the final structure were rotated and translated to desired positions (*Figure 1—figure supplement 1C*), and merged with two square flat bilayers of 26 × 26 nm$^2$ each, separated by 5 nm. The size of the bilayers was designed to provide space for the SNAREs to bend the membranes and induce fusion while limiting the overall size of the system.

*Four trans-SNARE complexes bridging two flat bilayers including four Ca$^{2+}$-bound C$_2$AB molecules* (Sqscff system). To generate C$_2$AB molecules for this simulation, the C$_2$AB molecule from a complex with the SNAREs (PDB accession number 5CCH) (*Zhou et al., 2015*) was used as a starting point. Ca$^{2+}$ ions were added at the corresponding sites as observed in the solution NMR structures of the C$_2$A and C$_2$B domains (PDB accession codes 1BYN and 1K5W) (*Fernandez et al., 2001*; *Shao et al., 1998*). After solvation with explicit water molecules and energy minimization, a 10 ns production MD simulation was carried out using position restraints to keep the initial coordinates of the C$_2$B domain and move the C$_2$A domain to a designed location so that the Ca$^{2+}$-binding loops of both C$_2$ domains point in similar directions (*Figure 1—figure supplement 1A*) and hence can bind in a Ca$^{2+}$-dependent manner to the same membrane (force constant 1,000 kJ/mol/nm$^2$). The final C$_2$AB structure was energy minimized and four copies of it were incorporated into the system containing four trans-SNARE complexes bridging two flat bilayers (qscff), interspersed between the SNARE complexes but without contacting them.

*Four trans-SNARE complexes bridging a vesicle and a flat bilayer* (qscv system). The vesicle was built by adaptation of the scripts for building coarse-grained vesicle systems from CHARMM-GUI Martini Maker (*Qi et al., 2015*). The radius of the vesicle was set to 11 nm and the number of lipids in the inner and outer layer of the vesicle, given the specific vesicle radius and lipid ratio in each layer, were calculated using the same scheme as in Martini Maker. As the final system was too large for long-time equilibration of the lipids in the inner and outer layer using water pores along the x, y and z axis, no water pore was created in the vesicle (i.e. water pore radius was set to 0 nm). The four trans-SNARE complexes were slightly modified with respect to those used for the qscff system to tilt the synaptobrevin TM regions such that they were perpendicular to the vesicle surface, and to tilt the SNARE four-helix bundles such that their long axis had similar angles with respect to the vesicle and the flat bilayer (*Figure 2B*, *Figure 2—figure supplement 1A*). The trans-SNARE complex built for the qscff system was used, after solvation, minimization and equilibration, as a starting point for a 2 ns production MD simulation imposing position restraints to keep all heavy atoms of the N-terminal half of the SNARE complex in their original designed coordinates and to force the backbone atoms of the TM regions of synaptobrevin and syntaxin-1 to move to their designed positions (force constant 1,000 kJ/mol/nm$^2$). Four copies of the final structure were rotated and translated to designed positions (*Figure 2—figure supplement 1A*), and merged with the vesicle and a square flat bilayer of 26 × 26 nm$^2$ each. After solvation and equilibration, we performed a 7 ns production MD simulation and observed the appearance of holes in the vesicle that arose because the lipid density was not optimal. The holes were filled manually with lipid patches from the original vesicle and a 5 ns production MD simulation was carried out with position restraints to keep the SNAREs in their initial locations. New holes appeared and were filled manually again. After another 10 ns production MD simulation with position restraints on the SNARE coordinates, no additional holes appeared. A final 80 ns production MD simulation with position restraints on the SNAREs was performed to equilibrate the vesicle lipids, which yielded the initial equilibrated system (*Figure 2—figure supplement 1B*,C) that was used to initiate an unrestrained production MD simulation.

*First simulation of primed complexes bridging a vesicle and a flat bilayer* (prsg system). The complexin-1 fragment [Cpx1(27-72)] was built starting from the crystal structure of Cpx1(26-83) bound to the SNARE complex (PDB accession code 1KIL) (*Chen et al., 2002*), which contained electron density for residues 32–73 of complexin-1. Five additional N-terminal residues that may be important for the steric clashes of complexin-1 with the vesicle, which were proposed to underlie the inhibitory activity of the accessory helix (*Trimbuch et al., 2014*), were added in a random conformation with Pymol. The initial conformation of the C$_2$AB fragment was generated starting from the coordinates of C$_2$AB used for the Sqscff system and, after solvation, minimization and equilibration, a 5 ns production MD simulation was performed with position restraints to keep the C$_2$B domain at its original position (force constant 1,000 kJ/mol/nm$^2$) and additional position restraints to move the C$_2$A domain so that its Ca$^{2+}$-binding loops can readily interact with the flat bilayer while the C$_2$B domain binds to the flat bilayer through the polybasic face (force constant 100 kJ/mol/nm$^2$).

To create the initial configurations of the primed complexes, we designated four positions to place the SNARE four-helix bundles bound to Cpx1(27-72) and to the Syt1 C$_2$AB through the primary interface as determined by crystallography (*Chen et al., 2002*; *Zhou et al., 2015*). The position of the C$_2$B domain was designed such that the polybasic region was placed close to but not contacting the flat bilayer to limit the bias introduced by the initial configurations of the primed complexes. The four trans-SNARE complexes generated for the qscv system were used as starting point to generate the four primed complexes. Four Cpx1(27-72) molecules were rotated and translated to interact each with a corresponding trans-SNARE complex based on the crystal structure (*Chen et al., 2002*), while four C$_2$AB molecules were placed at their final designed places. The system was solvated, minimized and equilibrated, and a 1 ns MD simulation was carried out with the following position restraints: (i) strong restraints (force constant 4000 kJ/mol/nm$^2$) to keep the synaptobrevin TM regions in their initial positions, as they were intended to remain inserted in the same positions in the vesicle; (ii) mild position restraints (force constant 100 kJ/mol/nm$^2$) to keep the C$_2$AB molecules at their initial (designed) positions; (iii) mild position restraints (force constant 100 kJ/mol/nm$^2$) on the syntaxin-1 TM regions to move them to their intended designed positions in the flat bilayer, which we planned to place further from the vesicle that in the qscv system; and (iv) mild position restraints (force constant 100 kJ/mol/nm$^2$) on the N-terminal half of the SNARE four-helix bundle and on Cpx1(27-72) to move

them to their designed positions so that each SNARE complex interacted with a corresponding $C_2AB$ molecule through the primary interface as in the crystal structure (*Zhou et al., 2015*). Note that we did not include any position restraints on the C-terminal half of the SNARE four-helix bundle and the juxtamembrane regions to allow them to adapt to the imposed restraints. We built a square flat bilayer of $31.5 \times 31.5$ nm$^2$ to provide sufficient space for interactions with the proteins and placed it 2.3 nm below the equilibrated vesicle from the qscv system. The flat bilayer and the vesicle were then merged with the four primed complexes to provide the starting point for the simulation.

*Second simulation of primed complexes bridging a vesicle and a flat bilayer* (prs2 system). The four primed complexes were generated using the initial primed complexes from the prsg system (*Figure 3—figure supplement 1*) and running a 50 ns MD simulation with the same restraints used to create the initial complexes. The four resulting primed complexes were merged with the equilibrated vesicle from the qscv system and with a flat bilayer of $35 \times 35$ nm$^2$ that was built slightly larger than that used for the prsg system to provide more space for protein-membrane interactions.

*Four $C_2AB$ molecules bound to $Ca^{2+}$ and to four trans-SNARE complexes bridging a vesicle and a flat bilayer* (prsncpxca system). The Cpx1(27-72) molecules were removed from the four initial primed complexes of the prsg system and five $Ca^{2+}$ ions were added to the corresponding binding sites of each $C_2AB$ molecule. The resulting complexes were merged with the equilibrated vesicle from the qscv system and the same flat bilayer of $35 \times 35$ nm$^2$ used for the prs2 system.

## MD simulations

After energy minimization, all systems were heated to 310 K over the course of a 1 ns MD simulation in the NVT ensemble and equilibrated for 1 ns in the NPT ensemble using isotropic Parrinello-Rahman pressure coupling (*Parrinello and Rahman, 1981*). NPT production MD simulations were performed for the times indicated in for each system using 2 fs steps, isotropic Parrinello-Rahman pressure coupling and a 1.1 nm cutoff for non-bonding interactions. All simulations were performed at 310 K except one simulation with the qscff system, which was performed at 325 K after a 310 K simulation. Nose-Hoover temperature coupling (*Hoover, 1985*) was used separately for three groups: (i) protein atoms plus $Ca^{2+}$ ions if present; (ii) lipid atoms; and (ii) water and KCL. Periodic boundary conditions were imposed with Particle Mesh Ewald (PME) (*Darden et al., 1993*) summation for long-range electrostatics. The speeds of the production simulations ran on Frontera at TACC are indicated in *Table 1*.

## Analysis of vesicle-flat bilayer contacts

Because the vesicle and the flat bilayers of the different systems contain large numbers of atoms, it was impractical to analyze vesicle-flat bilayer contacts through measurement of the distances between all atoms of the vesicle and all the atoms of the flat bilayer in multiple time frames of a trajectory. To limit the calculations, we selected only oxygen atoms from frames taken at 1 ns steps of each trajectory and measured the distances between all the oxygen atoms of the vesicle and all the oxygen atoms of the flat bilayer in each frame. The number of vesicle-flat bilayer contacts in each frame was defined as the number of oxygen-oxygen distances below 1 nm.

## Acknowledgements

Most of the work presented in this paper was performed through a Pathways allocation for high performance computing using Frontera (project MCB20033) at the Texas Advanced Computing Center (TACC) at The University of Texas at Austin (URL: http://www.tacc.utexas.edu). This research also used computational resources provided by the BioHPC supercomputing facility located in the Lyda Hill Department of Bioinformatics, UT Southwestern Medical Center, TX (URL: https://portal. biohpc.swmed.edu). This work was supported by grant I-1304 from the Welch Foundation (to JR), by NIH Research Project Award R35 NS097333 (to JR), by NSF Research Project Award MCB-2111728 (to WI), and by the Natural Science Foundation of Shanghai Grant 19ZR1473600 (to YQ).

## Additional information

### Funding

| Funder | Grant reference number | Author |
| --- | --- | --- |
| National Institute of Neurological Disorders and Stroke | R35 NS097333 | Josep Rizo |
| Welch Foundation | I-1304 | Josep Rizo |
| National Science Foundation | MCB-2111728 | Wonpil Im |
| Natural Science Foundation of Shanghai | 19ZR1473600 | Yife Qi |
| University of Texas at Austin | | Josep Rizo |

The funders had no role in study design, data collection and interpretation, or the decision to submit the work for publication.

### Author contributions

Josep Rizo, Conceptualization, Formal analysis, Funding acquisition, Investigation, Methodology, Resources, Validation, Visualization, Writing – original draft; Levent Sari, Conceptualization, Formal analysis, Methodology, Validation, Visualization, Writing – review and editing; Yife Qi, Funding acquisition, Investigation, Methodology, Writing – review and editing; Wonpil Im, Funding acquisition, Methodology, Writing – review and editing; Milo M Lin, Conceptualization, Methodology, Validation, Visualization, Writing – review and editing

### Author ORCIDs

Josep Rizo (iD) http://orcid.org/0000-0003-1773-8311
Wonpil Im (iD) http://orcid.org/0000-0001-5642-6041
Milo M Lin (iD) http://orcid.org/0000-0001-8680-2685

### Decision letter and Author response

Decision letter https://doi.org/10.7554/eLife.76356.sa1
Author response https://doi.org/10.7554/eLife.76356.sa2

## Additional files

### Supplementary files

• Transparent reporting form

### Data availability

Most files corresponding to our molecular dynamics simulations are available in the dryad database (doi:10.5061/dryad.ns1rn8pw6). Because of the very large size of trajectory files, it was not practical to deposit them in this database, but these files are available from the corresponding author upon reasonable request.

The following dataset was generated:

| Author(s) | Year | Dataset title | Dataset URL | Database and Identifier |
| --- | --- | --- | --- | --- |
| Rizo J | 2022 | Data from: All-atom molecular dynamics simulations of Synaptotagmin-SNARE-complexin complexes bridging a vesicle and a flat lipid bilayer | http://doi.org/10.5061/dryad.ns1rn8pw6 | Dryad Digital Repository, 10.5061/dryad.ns1rn8pw6 |

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
