## [Editor Report]

Using all-atom molecular dynamics simulations to visualize the pre-fusion primed state during synaptic vesicle fusion is very original and this approach will certainly be used by others in the future. This work provides new insights into the protein organization prior to vesicle fusion that will help better understand the mechanisms of vesicle priming and evoked-release.

---

## [Decision Letter]

**Decision letter after peer review:**

Thank you for submitting your article "All-atom molecular dynamics simulations of synaptic vesicle fusion I: a glimpse at the primed state" for consideration by *eLife*. Your article has been reviewed by 3 peer reviewers, and the evaluation has been overseen by a Reviewing Editor and Vivek Malhotra as the Senior Editor. The following individual involved in the review of your submission has agreed to reveal their identity: Ben O'Shaughnessey (Reviewer #2).

Essential revisions:

1) Manuscript readability

a. the manuscript should be shortened. This notably applies to the Results section.

b. the manuscript should better focus on the relevant conclusions relative to the SNAREs (degree of zippering, juxtamembrane linker – lipid interactions), synaptotagmin and complexin. These modifications will be an opportunity to discuss the results in view of other simulations, i.e. coarse grain approaches accessing fusion timescales.

2) Limitations of all atom MD simulations

a. At the molecular level, the assumptions regarding the initial arrangement of the proteins and the missing aspects (e.g. Munc13, Munc18, Syt linkers, Cpx-SNARE interactions) must be explicitly stated and discussed in view of the current knowledge in the field.

b. Realistic statements about likely fusion times need to be compared to the all-atom simulation times.

3) "Primed" state

a. The term "primed", used in the title and in the manuscript is misleading because other core synaptic proteins are not included in the simulations.

b. It is difficult to assess whether the vesicle in the proposed molecular arrangement is actually primed. On the contrary, given the narrow intermembrane distance with molecular contacts, it is very likely the membranes will ultimately fuse. All-atom simulations cannot reach the relevant time scales to be conclusive.

c. The Cpx accessory helix looks like a wobbly finger unlikely to support much force.

4) Calcium addition

a. Calcium-phospholipid may play an important part in the molecular arrangement and the fusion process. This is ignored here and should be addressed.

b. How does one reconcile that the aliphatic loops on Synaptotagmin C2B domain do not insert into the membrane upon calcium binding as observed in previous structural/functional studies?

*Reviewer #1 (Recommendations for the authors):*

1) It is difficult to assess if the results from simulations are a true representative of the biological process or an outcome of the initial condition/constraints chosen. For example, it is puzzling that there is no intra-molecular assembly of the SNAREpins during the simulation even though the coil-coil interactions are expected to occur in the simulation time scales. It appears, in almost all cases, that the SNARE zippering is unaltered at the end of the simulation. Also, it might be possible that the authors' choice to model the juxtamembrane region as a fully unstructured region prevents membrane fusion under the current simulation conditions. While it is not clear if the SNAREs zippering extends through the juxtamembrane (JM) region into the transmembrane region as observed in the crystal structure of full-length SNARE (Stein et al. Nature 2009), it stands to reason the JM region needed to be at least partially structured for effective force transfer to catalyze merging of the bilayers.

2) A major conclusion of the report is that the steric clash between Complexin accessory helix and vesicle serves as the fusion clamp and indeed drives the positioning of the SNARE and Synaptotagmin on the planar bilayer. However, there are a couple of factors that might alleviate or even mitigate this steric clash: (i) the vesicle and bilayer are positioned at ~2.3 nm apart at the beginning of the simulation. However, high-resolution cyroEM analysis in synaptosomes/cultured neurons (Fernandez-Busnadiego, R. J Cell Biol 2013; Radhakrishnan et al. PNAS 2021) show that the inter-bilayer distance of docked/primed vesicle is ~4.5 nm. Thus, it might be imperative to carry out the simulation with the physiological accurate inter-bilayer distance (ii) Complexin molecule has been positioned on SNAREs assuming a fully-zippered SNARE complex. However, there is sufficient evidence that SNAREs are likely only partially-assembled in an RPP vesicle (Hua & Charlton, Nat Neurosci 1999, Prashad & Charlton, PLoS One, 2014), and the positioning of the CPX, esp. the accessory helix is correlated to the extent of SNARE assembly (Choi et al. *ELife* 2016, Kummel et al. Nat Struct Biol 2011; Zhou et al. Nature 2017). Furthermore, accessory helix has been shown to interact with c-terminal ends of t- and v-SNARE molecules (Kummel et al. Nat Struct Mol Biol 2011; Malsam et al., Cell Reports 2020). Thus, it is possible that the alternate positioning of the accessory helix and other interactions might reduce the observed steric clash.

3) How does one reconcile that the aliphatic loops on Synaptotagmin C2B domain do not insert into the membrane upon calcium binding as observed in previous structural/functional studies (Grushin et al. Nat Comms 2019; Kuo et al. J Mol Biol 2009) even though synaptotagmin interacts with the membrane, including partial insertion of the C2B aliphatic loop, under calcium-free conditions. This is a rather crucial and missing piece considering that calcium-triggered membrane insertion is predicted to be the driving force for triggered fusion.

*Reviewer #2 (Recommendations for the authors):*

1. The authors' major conclusion is that the AA simulations support the model of Voleti et al. for the organization that clamps fusion in the pre-ca primed state. However, from Figures 3, 4 (and associated figure supplements) fusion seems very likely not to be clamped, given the vesicle contacts the planar membrane (the degree of contact is still growing at the end of the simulation, Figure 4 supp 4). As stated in (lines 420-425) the vesicle membrane is not flattened. This indicates a lower force than with SNAREs alone, but seems unlikely to block fusion. Due to running time limitations, AA simulations cannot test if fusion would occur in a physiological time. The structure does not keep the membranes apart, as it rotates and permits contact. The authors are clear about this – indeed, to predict the orientation is stated as a major objective. But the conclusions of lines 365-367 and the final sentence of the abstract, suggesting these results demonstrate a fusion clamp, seems unjustified as far as I can see. The emphasis on the cpx accessory helix role also appears somewhat exaggerated, as if on its own it provides a mini-buttress that separates vesicle and planar membrane. It's hard for me to imagine it supports much force in this configuration.

2. The simulations with bound calcium (final section of Results) seem inconclusive. The number of contacts is still growing at the end of the simulation, and we cannot know if the C2B will ever dissociate from the SNAREs. It's very reasonable to try this simulation but given the outcome I'm not sure a long section is merited, particularly with the tentative title "Potential effects of Ca2plus binding to synaptotagmin-1." This negative, albeit interesting finding, might be briefly summarized in the main text.

3. The manuscript would be strengthened by a more balanced presentation acknowledging the limitations of AA simulations (while of course still extolling their merits) and connecting to some degree with analysis on other scales, including coarse-grained approaches beyond MARTINI. SNARE-mediated fusion was studied using ultra coarse-grained (Mostafavi et al., 2017; McDargh et al., 2018) and even continuum (Manca et al., 2019) representations. Every approach has strengths and weaknesses. AA approaches scrutinize local issues as no others can, but presently they are remote from being able to demonstrate hemifusion, fusion, unclamping and ca-evoked fusion. Making matters worse, NT release is clearly stochastic, so multiple runs are needed for each condition. These limitations are apparent in this study: almost every conclusion comes with a caveat related to running time. In previous seminal MARTINI studies that achieved fusion (Risselada, Sharma and Lindau) the conditions were intentionally biased for fusion (vesicle size, lipid composition, temperature, helical LDs) or nanodiscs were used. In (Risselada, 2011) no fusion was observed when the LD was made unstructured.

In the Introduction the authors assert that experiments suggest "..the fusion step occurs in just a few microseconds," which timescales AA simulations may be able to access. They quote the 60 microsec delay times (ca influx to first sign of the excitatory post synaptic current, EPSC) reported by Sabatini and Regehr at 38 degrees. However, 0.5 -2 ms is much more typical in the literature (admittedly, the 38 degrees study is distinguished by the temperature being physiological.) Related, long-ago Katz argued other processes (e.g. NT diffusion across the synaptic cleft) are much faster than NT release (Katz and Miledi, 1965).

4. In simulations with SNAREs only, the SNARE complexes are cleared laterally, and the membranes are squashed together, generating an ECZ (extended contact zone, a flat portion of vesicle), Figures 1C, 1F. This is precisely the behavior seen in highly coarse-grained simulations (Mostafavi et al., 2017, Mcdargh et al., 2018), where entropic SNARE-SNARE and SNARE-membrane forces cleared the fusion site and pressed the vesicles together (those studies used undeformable membrane surfaces, so no vesicle flattening occurred). The entropic forces were predicted to provoke fusion after a time of order msec, with faster fusion for more SNAREs. These coarse-grained simulations and their relation to the present findings should be discussed.

The authors suggest the pressing together of the membranes is caused by binding of the LDs to the vesicle membrane (lines 262-266). This does not seem a plausible alternative to the proposed entropic forces, as LD-membrane adhesion would not favor the SNAREs being pushed outwards as far as I can see.

The authors argue that the ECZ in the SNARE-only simulations suggests SNAREs alone cannot fuse membranes rapidly, since fusion was slow in Hernandez et al., 2012 and Witkowska et al., 2021 where ECZs were seen. However, in those in vitro studies many other processes preceded fusion (SNARE assembly, docking etc) and micron scale GUVs were used by Witkowska et al.

5. A concern is the presentation, whose clarity would benefit from a more concise text. It is laudable to convey the details (computational papers where readers cannot tell what was done are frustrating), but many passages are long repeats of previous passages. For example, opening paragraphs of sections in Results often repeat descriptions of simulations in previous sections at great length, then specifying what was different in the current section. These could be massively shortcut. Short summaries in the main text, with details left to Methods or Supplementary materials, would be more digestible for readers.

6. This paper describes many current hot issues in the field, a great service. The figures are very nice but would be helped by a simple visual key to identify β sandwiches, the polybasic face, ca-binding loops, etc. For an uninitiated reader, it is tough staring at these protein structures trying to figure out which features are where? Also, I suggest adding a length bar to one or more Figures

7. It is stated that the Cpx accessory helix inhibits release "likely" because it causes steric clashes with the vesicle (line 85). I think this is a powerful and very reasonable suggestion, but perhaps "possibly" would better reflect current uncertainty about the mechanism.

8. Their 26 nm diameter vesicles are ~ 2-fold smaller than synaptic vesicles. I do understand why this measure is taken (and the authors mention why), but the synaptic vesicle size should be stated.

*Reviewer #3 (Recommendations for the authors):*

My overall sense from this study is that the simulation efforts are preliminary and sufficiently incomplete to cause concern about the validity of the conclusions. I am concerned about several omissions and their potential impact on conclusions about the prefusion complex and the possible trajectories leading to fusion:

1. SNARE/Syt/Cpx omissions – What is the potential impact of removing the Habc region of syntaxin 1 given its significant excluded volume and potential interactions with membrane PIP2? Similarly, excluding the palmitoylated linker regions of SNAP-25 may play important and interesting roles affecting SNARE orientation, the distribution of forces between SNAREs and membranes, and membrane behavior. The lack of a Syt1 juxtamembrane region (as well as its transmembrane anchor) seems like a real missed opportunity given past work suggesting several interesting hypotheses for intramolecular and membrane interactions of this region. Finally, omitting the C-terminal domain of Cpx1 with its known membrane-interacting region may have significant implications for the detailed behavior of Cpx1 and the forces acting on its SNARE-binding region. While no realistic simulation could currently hope to capture all of this, I would have preferred fewer simulations with more assessment of whether or not some of these omissions would cause major changes to the behavior of the simulated system.

2. Calcium-phospholipid interactions – When the authors included 5 calcium ions per Syt1 to assess the impact of elevated local calcium on the simulation dynamics, I was struck by a lack of corresponding calcium interactions with PS and PIP2. 20 calcium ions in the simulated volume would roughly correspond to 1 mM calcium, and even that wouldn't necessarily lead to all 20 potential binding sites on Syt1 being occupied. At the same time, one would expect divalent interactions with PS and PIP2, which could neutralize membrane repulsion and significantly lower at least one aspect of the complex membrane fusion energy barrier. Work by chemists such as Feigenson have indicated strong calcium-mediated interactions between even PS and PC at concentrations much lower than 1 mM (Biochemistry 1989). Some of these chemical details may not be capable of proper simulation in the MD formalism deployed in the current study, but this should be addressable in some fashion.

3. I was not convinced by the authors' reasoning regarding one microsecond being a relevant timescale for synaptic vesicle fusion. And given that even some initial phase of membrane fusion was not observed in these simulations, I find it impossible to access wherein the process of priming/fusion these current simulations reside. The fastest reported latency between presynaptic calcium entry and fusion is around 60 microseconds as the authors point with the Sabatini/Regehr study. Importantly, that was not a single-synapse measurement but instead, a population measure involving 1000s of synapses. So the first latency likely represents a small population from the fast tail of a distribution of fusion times. And given the 1-2 microsecond delay for cleft glutamate diffusion and the 10-20 microsecond activation time of a stellate cell AMPA receptor, it is likely that the calcium-fusion delay at this synapse resides in the 50-100 microsecond time window. Thus, a 400-nanosecond simulation would seem far too brief to do this process justice.

4. Since the simulations are certainly not trying to capture relevant roles and impacts of other core synaptic proteins such as Munc13 and Munc18, I thought that the use of 'primed' state was a bit oversold and misleading in this manuscript. These simulations seem most appropriate for interpreting in vitro liposome fusion experiments utilizing just SNAREs or SNAREs plus Cpx/Syt1. I am not sure what it would mean to describe a primed state for the SNAREs and synaptic vesicle without also having Munc13 present and bound at least to the two membranes if not also to the SNAREs. I appreciate that the authors are modeling something that represents our best guess for the SNARE assembly on a tightly docked and primed vesicle, but this simulation clearly lacks crucial elements that go into what the field usually refers to as a primed synaptic vesicle. I would want the language used to reflect this as much as possible.

5. I do not have a sense for how worrisome it is from a technical perspective to forgo replicate simulations. For instance, is it better to have two replicates each of three simulations rather than six slightly different simulations each done once? It would be useful to have some discussion of the uncertainty/reliability attached to these conclusions given the absence of replicates.

[Editors' note: further revisions were suggested prior to acceptance, as described below.]

Thank you for resubmitting your work entitled "A glimpse at the primed Synaptotagmin-SNARE-complexin complex from all-atom molecular dynamics simulations" for further consideration by *eLife*. Your revised article has been evaluated by Vivek Malhotra (Senior Editor) and a Reviewing Editor.

The manuscript has been improved but there are some remaining issues that need to be addressed, as outlined below.

Please address the points brought up by Reviewer #3 on the primed state and on calcium/lipid interactions, at least at the writing level. To avoid ambiguity, it would be better to remove the word 'primed' from the title, lines 34 and 37 in the abstract, and line 129 in the introduction, and change the corresponding sentences when needed. This is probably not sufficient because there are so many mentions of 'primed states', primed complexes', or 'vesicle priming'.

Regarding the interactions of calcium with lipids, we realize that getting computational time is a limiting and costly resource currently. Asking to perform additional simulations involving lipid/calcium interactions may be difficult but the authors can certainly mention that it is a limitation of their simulations that may affect the outcome and should be tested in the future.

*Reviewer #3 (Recommendations for the authors):*

Rizo and colleagues have shortened and edited their manuscript as requested in the first review. I wasn't strongly enthusiastic about this MD study during the first round and remain somewhat dissatisfied after reading the authors' response to our concerns. Two of my concerns were largely ignored by the authors in their rebuttal but remain worrisome to me nonetheless.

One concern is the authors' continued declaration that their simulations are synonymous with the primed fusion complex. Vesicle priming is already a somewhat muddled concept in the field and this manuscript doesn't help the confusion. I appreciate that they edited their title a bit but anyone glancing at the paper or searching for it on PubMed would very likely interpret this as the primed state prior to fusion. In addition, they conclude in the abstract (line 37) that 'the primed state contains macromolecular assemblies …' whereas I don't believe the simulations warrant this conclusion. This is reiterated in the last sentence of the introduction (line 129-132) but at least they soften the conclusion with 'suggest that'. I am not sure the authors got that much more out of the model than they put in to begin with since they chose starting points that they were already convinced represented their best guess at the primed state of the fusion complex. Perhaps some of the observations regarding the juxtamembrane linkers of the SNAREs are moderately unexpected, but given that no fusion was witnessed, the reader doesn't know which details of the current model truly correspond to relevant prefusion scenarios.

My other concern is that the authors continue to ignore the very real possibility that calcium interactions directly with the phospholipids (independent of Syt1 C2 domains) are a critical aspect of membrane fusion. This has been studied chemically and using in vitro membrane fusion assays for 50 years but wasn't even discussed as a possible explanation for the lack of fusion in the simulation where calcium was included. Just to be explicit, I am thinking of papers such as Papahadjopoulos BBA 1976, Feigenson Biochem 1986,1987, and 1989 studies, Kachar Biophys J 1986, and modern studies such as Churchward Biophys J 2008. While I don't know what the technical limitations of implementing calcium-phospholipid interactions are in all-atom MD, I can find examples in the literature such as Allolio and Harries ACS Nano 2021 and Allolio et al. PNAS 2018 where calcium ion interactions with phospholipids during membrane fusion are explicitly incorporated, so I assume there isn't a fundamental reason this cannot be explored or acknowledged. I don't think it would be surprising if some of the key results here such as the juxtamembrane linker electrostatic interactions with the membrane would be strongly affected in addition to the possibility that the fusion energy barrier would be lowered sufficiently to witness the beginning of a fusion event on a microsecond time scale.

Overall, this was a nice first effort at an ambitious simulation scale and could serve as an introductory template for future attempts at modeling SNARE-mediated fusion. The preliminary and underdeveloped feel of the manuscript and notably, the lack of some sort of fusion-like transition captured in the simulations diminish my enthusiasm a bit.

---

## [Author Response]

General comments:

We thank the reviewers for their very careful review of our manuscript and their constructive criticisms, which have helped us to substantially improve the quality of the manuscript. We particularly appreciate the positive overall evaluation despite the concerns regarding the limitations of our simulations.

On a personal note, I would like to point out that I was very aware of these limitations when I initiated this project. Nevertheless, I decided to devote a very large amount of my own time to perform these simulations with the help from my co-authors because, after 29 years working intensely in this field, I felt that we really did not understand membrane fusion and it was very difficult to obtain a description of the dynamic events that lead to membrane fusion at atomic detail through experiments only. I believed that there was a chance that we might observe initiation of membrane fusion if we came up with a proper initial configuration. Just seeing some lipids emerging from the bilayers to initiate formation of non-bilayer intermediates that are necessary for membrane merger would be tremendously informative and would lead to a hypothesis(es) that could be tested experimentally. And, even if we did not observe initiation of fusion, we still could learn important insights. Moreover, the three-dimensional models that we built to perform the simulations would be very valuable tools to visualize this system and evaluate mechanistic models. I believe that the results described in the paper support these expectations.

The simulations were performed at TACC on Frontera, one of the fastest supercomputers available for academic research, with a Pathways allocation that I was awarded through XSEDE. The allocation provided 246,940 node hours and the simulations of the larger systems were performed using 32-48 nodes. To perform additional simulations to address some of the reviewer concerns would require application for another large allocation specifically targeted for this purpose and, if the allocation was awarded, it would take several months to perform the simulations. Hence, performance of these simulations would unduly delay publication of our results, which we believe will already be of interest to the research community. Moreover, this paper provides a framework not only for my lab but also for others to perform additional all-atom simulations, which provide an important and timely complement to experimental and coarse-grained simulation data. We hope that the systems that we describe will serve as a basis to apply for high performance computing time and continue this research in labs around the world.

The reviewers will see that we have different opinions on some of the responses described below, which often arise from different appreciation of the available experimental data. We hope the reviewers realize that some concepts are much less well established than is often stated in the literature and will keep an open mind about some of the specific issues that we debate.

Essential revisions:1) Manuscript readabilitya. the manuscript should be shortened. This notably applies to the Results section.

The manuscript has been considerably shortened and in particular the Results section, which is now five pages shorter.

b. the manuscript should better focus on the relevant conclusions relative to the SNAREs (degree of zippering, juxtamembrane linker – lipid interactions), synaptotagmin and complexin. These modifications will be an opportunity to discuss the results in view of other simulations, i.e. coarse grain approaches accessing fusion timescales.

We have removed the last simulation presented in the original manuscript, which was designed to investigate how the synaptotagmin-1 C_2_ domains cooperate with the SNAREs in triggering fusion of a vesicle and a flat bilayer, because it did not offer conclusive insights about fusion. We have also rewritten most of the discussion to focus on the most relevant conclusions as well as to discuss several key issues raised by the reviewers. We realize that, although we were cautious in presenting our conclusions, we did not discuss alternative views sufficiently. The reviewer comments have been very helpful to provide a more balanced perspective. Moreover, in the introduction and the discussion we now emphasize the importance of continuum and coarse-grained approaches as a complementary strategies to all-atom simulations, while also providing a more concrete justification for the need for atomistic simulations:

(lines 98-112): ‘Simulations using continuum and/or coarse-grained representations have provided important insights into SNARE-mediated membrane fusion (Fortoul et al., 2015; Kasson et al., 2006; Manca et al., 2019; McDargh et al., 2018; Mostafavi et al., 2017; Risselada et al., 2011; Sharma and Lindau, 2018). Continuum models can access the longest timescales, but require experimental data or atomistic simulations to parameterize the material properties, and often need to constrain geometries or material properties due to lack of context-dependent parameters (Fortoul et al., 2015). Coarsegrained molecular simulation approaches are freed from some of these constraints but at the expense of reduced simulation speed, and are limited in their ability to capture certain entropic effects and protein conformational changes (see below). To date, coarse-grained models of SNARE-mediated fusion have accessed the low microsecond timescale (Kasson et al., 2006; Risselada et al., 2011; Sharma and Lindau, 2018). All-atom simulations are better suited to reproduce the finely-balanced network of interactions between proteins, ca^2+^ and lipids that are expected to lead to membrane fusion but, because of the large size of the systems involved (millions of atoms), the low microsecond time scale has only recently become accessible to the most powerful available high-performance computing resources.’

(lines 593-599): ‘Continuum and coarse-grained simulations, which have already provided important insights into SNARE-mediated membrane fusion (Fortoul et al., 2015; Manca et al., 2019; McDargh et al., 2018; Mostafavi et al., 2017; Risselada et al., 2011; Sharma and Lindau, 2018), offer the opportunity to explore much longer time scales with similar systems. Hence, a marriage of approaches whereby the most interesting results obtained by such simulations are investigated in further detail by all-atom simulations will likely provide a powerful strategy to unravel the intricate mechanisms that govern fast ca^2+^-triggered membrane fusion.’

2) Limitations of all atom MD simulationsa. At the molecular level, the assumptions regarding the initial arrangement of the proteins and the missing aspects (e.g. Munc13, Munc18, Syt linkers, Cpx-SNARE interactions) must be explicitly stated and discussed in view of the current knowledge in the field.

We now point out the limitations of the simulations, including the absence of key components, in the abstract, at the end of the introduction and in the discussion:

(lines 32-33): ‘Our results need to be interpreted with caution because of the limited simulation times and the absence of key components’.

(lines 123-124): ‘Because of the limited simulation times and the absence of key components, our results cannot lead to definitive conclusions’

(lines 436-438): ‘Our results need to be interpreted with caution because of the limited simulation times, the dependence of the results on the initial configurations and the absence of key elements of the release machinery’

(lines 581-591): ‘Clearly, these models will need to be tested with further MD simulations and experimentation, and there are multiple factors of the primed state of synaptic vesicles that were not included in the simulations. As mentioned above, particularly important for this state is Munc13-1, but other missing elements include Munc18-1, the N-terminal region of syntaxin-1, the SNAP-25 linker joining its SNARE motifs, the N- and C-terminal regions of complexin-1 and the TM region plus linker sequence of Syt1 [reviewed in (Rizo, 2022)]. Moreover, the Syt1 C_2_B domain can also bind to the SNARE complex through a so-called tripartite interface whereby an α-helix of the C_2_B domain is adjacent to the complexin-1 helix (Zhou et al., 2017), which could stabilize the complexin1-SNARE interface to hinder rotations that might lead to dissociation (see Figure 3K, Figure 3—figure supplement 2F). Moreover, there is evidence that Syt1 forms oligomeric rings that hinder spontaneous neurotransmitter release (Tagliatti et al., 2020). It is also plausible that the SNAREs alone can induce fast fusion in different configurations that we did not study.’

We hope the reviewers realize that many important results in our field were obtained with experiments using minimal systems that included even fewer components than our simulations, and those contributions are now widely considered as seminal. Moreover, the systems that we built provide a framework to add additional components in the future.

b. Realistic statements about likely fusion times need to be compared to the all-atom simulation times.

We hope that the reviewers can keep an open mind about this issue. We have tried to improve the rationale behind the statement that the fusion step may occur in a few microseconds in the introduction:

(lines 112-120): ‘In this context, it is worth noting that the delay from ca^2+^ influx into the presynaptic terminal to observation of postsynaptic currents in rat cerebellar synapses at 38°C is 60 µs (Sabatini and Regehr, 1996), and that multiple events occur within this time frame, including ca^2+^ binding to the sensor, release of inhibitory interactions that hinder premature fusion, ca^2+^-evoked synaptic vesicle fusion, opening of the fusion pore, diffusion of neurotransmitters through the synaptic cleft, binding of the neurotransmitters to their postsynaptic receptors and opening of the channels that underlie the postsynaptic currents. These observations suggest that the fusion step may occur in just a few microseconds and hence that it may be possible to recapitulate the initiation of ca^2+^-dependent synaptic vesicle fusions in all-atom MD simulations starting with a properly designed initial configuration.’

Please note that we do not state definitively that the fusion step occurs in just a few microseconds; we present this as a possibility. And we now do not suggest the possibility of recapitulating the entire process of fusion with all-atom MD simulations, but we do suggest that it may be possible to observe the initiation of fusion. It is also worth noting that folding of small proteins can occur in the low microsecond time scale, so this time scale might allow substantial rearrangements necessary to initiate fusion, and that the available coarse-grained simulations of SNARE-mediated fusion were performed in the 1 µs time scale (Kasson et al., 2006; Risselada et al., 2011; Sharma and Lindau, 2018).

We also would like to emphasize that the rate limiting step may not be fusion itself but another of the multiple events that occur from ca^2+^ influx to postsynaptic currents, particularly the release of inhibitory interactions that hinder premature fusion. For instance, the interaction of the synaptotagmin-1 C_2_B domain with the SNARE complex is believed to hinder neurotransmitter release, which is supported by screens for mutations that remove dominant negative phenotypes of synaptotagmin-1 mutants [Guan et al. (2017) *ELife* 6, e28409] and by the finding that a mutation that enhances this interaction (E295A/Y338W) impairs release [Zhou et al. (2015) Nature 525, 62; Voleti et al. (2020) *eLife* 9, e57154]. The dissociation constant of this interaction is estimated to be about 20 µM, which translates to an off rate of 2,000 Hz (0.5 ms time scale) assuming that the on rate is diffusion limited. Dissociation is likely accelerated by a ca^2+^-induced change in the orientation of the C_2_B domain with respect to the membrane, but we do not know to what extent. Hence, we really do not know what is the time scale of the fusion step itself. We now discuss these issues in the following sentences:

(lines 388-392): ‘Based on the estimated k_D_ of the interaction between the C_2_B domain and the SNARE complex [ca. 20 µM (Voleti et al., 2020)], the off rate for dissociation is expected to be at most 2,000 Hz and hence too slow for the time scales reachable in our simulations. However, it is plausible that dissociation might be strongly accelerated by changes in the orientation of the C_2_B domain with respect to the membrane induced by ca^2+^ (Voleti et al., 2020)’.

3) "Primed" statea. The term "primed", used in the title and in the manuscript is misleading because other core synaptic proteins are not included in the simulations.

We agree that the term ‘primed state’ in the title may be misleading and have replaced this term in the title with ‘primed Synaptotagmin-SNARE-complexin complex’. We hope that this term will be acceptable to the reviewers, as the notion that a complex of synaptotagmin, SNAREs and complexin is a central element of the primed state is widely accepted in the literature [e.g. Zhou et al. (2017). The primed SNARE-complexin-synaptotagmin complex for neuronal exocytosis. Nature 548, 420]. Please note that we use the word ‘glimpse’ in the title to soften the message and not give the impression that we have elucidated the structure of the primed complex. Furthermore, as stated above, we now point out in prominent parts of the manuscript that key components are absent in our simulations.

b. It is difficult to assess whether the vesicle in the proposed molecular arrangement is actually primed. On the contrary, given the narrow intermembrane distance with molecular contacts, it is very likely the membranes will ultimately fuse. All-atom simulations cannot reach the relevant time scales to be conclusive.

We agree that we cannot be sure that the proposed molecular arrangement reflects that present in the primed state. This is why we are careful throughout the text to use terms like ‘suggest’ or ‘indicate’ rather than ‘show’ or ‘demonstrate’. We hope that these terms are acceptable, as we must be able to somehow describe what the simulations are telling. We also note that we place particular emphasis on results from our simulations that have clear correlations with experimental data. For instance, the physiological relevance of the interactions of the SNARE complex with complexin and with the primary interface of the synaptotagmin-1 C_2_B domain are supported by overwhelming experimental evidence [e.g. Xue et al. (2007) Nat Struct Mol Biol 14, 949; Maximov et al. (2009) Science 323, 516; Zhou et al. (2015) Nature 525, 62; Guan et al. (2017) *ELife* 6, e28409], and these interactions are generally believed to occur in the primed state given the role of synaptotagmin-1 and complexins in the ca^2+^-triggered step of neurotransmitter release.

We also would like to emphasize that there was a total of eight synaptotagmin-1-SNAREcomplexin-1 assemblies in the two simulations of the primed complex, and the arrangements observed at the end of the simulations were very similar in all of them. The observed arrangements make a lot of sense from the chemical-biophysical point of view, as they allow very extensive interactions of the polybasic face of the C_2_B domain with the lipids while maintaining the C_2_B domain-SNARE and complexin-1-SNARE interfaces observed by crystallography. The physiological importance of some residues of the polybasic face has also been well established [e.g. Brewer et al. (2015) Nat Struct Mol Biol 22, 555]. We now discuss these issues in lines 484-499.

We agree that membranes that are in contact will eventually fuse, but the key point is: at what rate? We now discuss this issue in the following sentences:

(lines 471-478): ‘These extended interfaces have been observed by cryo-EM and by fluorescence microscopy, and evolve to fusion in longer time scales (seconds-minutes) (Diao et al., 2012; Hernandez et al., 2012; Witkowska et al., 2021), in agreement with data showing that liposome fusion occurs minutes after liposome docking (Cypionka et al., 2009). SNARE-mediated fusion was also observed at faster time scales (Domanska et al., 2009; Heo et al., 2021). Thus, it is plausible that fusion is slower under conditions that favor formation of extended interfaces. Interestingly, cryo-EM studies indicated that formation of such extended interfaces is hindered by other proteins involved in ca^2+^evoked release, which favor point-of-contact interfaces that fuse faster (Diao et al., 2012; Gipson et al., 2017).’

Note that in the cryo-EM analyses of Hernandez et al. the extended contact interfaces between liposomes occur early in the reactions and gradually disappear, and that Diao et al. also conclude that extended interfaces fuse slowly. Moreover, spontaneous release of synaptic vesicles occurs in the minute time scale at hippocampal synapses [e.g. Rhee et al. (2005) PNAS 102, 18664; Xue et al. (2008) PNAS 105, 7875]. Hence, the notion that primed synaptic vesicles may be in contact with the plasma membrane is not inconsistent with the observed spontaneous release rates.

Nevertheless, we agree that the membranes may be further apart in the primed state and in the discussion now explain that the arrangement of the primed complexes that we observed can be consistent with a range of distances:

(lines 572-580): ‘In the second scenario, the distance from the vesicle to the plasma membrane in the tight primed state is a few nm. Manually moving the vesicle in the pose of Figure 7B to a distance of 3 nm from the flat bilayer (Figure 7D) shows that the same spring-loaded configuration of the Syt1 C_2_B domain-SNARE-Cpx1(27-72)-flat bilayer assembly could be kept by stretching the synaptobrevin juxtamembrane linker, and the Cpx1(27-72) helix would still hinder progress toward final zippering and fusion. This configuration is also largely compatible with longer vesicle-flat bilayer distances if the linker is stretched further and/or there is partial SNARE unzippering. In this case, steric clashes of the Cpx1(27-72) helix with the vesicle might be alleviated or eliminated, but they would occur as soon there is full zippering and hence would still hinder vesicle fusion.'

With regard to the comment that ‘All-atom simulations cannot reach the relevant time scales to be conclusive’, please see the response to point 2b above.

c. The Cpx accessory helix looks like a wobbly finger unlikely to support much force.

We invite the reviewers to think about these mechanistic issues in terms of energy barriers that slow down reactions. We now discuss this issue in energetic terms:

(lines 525-531): ‘It is difficult to estimate the energy barrier imposed by this behavior on SNARE zippering, but it is known that the energy cost of breaking just one hydrogen bond to distort the complexin-1 helix can be about 1.8 k_B_T (Nick Pace et al., 2014), which could translate to 7.2 k_B_T by cooperative action of four primed complexes. Such an energy barrier would slow down SNARE zippering by a factor of 1,340 (=e^7.2). For comparison, the enhancements of spontaneous neurotransmitter release observed upon deletion of complexin range from 3 to > 20 (Huntwork and Littleton, 2007; Martin et al., 2011). Thus, even a small energy barrier caused by steric hindrance can account for these enhancements.’

It is also important to consider that there may be other energy barriers, and the acceleration of release induced by ca^2+^ is expected to arise in substantial part from ca^2+^dependent interactions of the synaptotagmin-1 ca^2+^-binding loops with the lipids.

4) Calcium additiona. Calcium-phospholipid may play an important part in the molecular arrangement and the fusion process. This is ignored here and should be addressed.

This issue was not ignored. In the introduction we clearly stated that ca^2+^-dependent binding to phospholipids is critical to trigger release (now in lines 66-68 of the revised manuscript). Ca^2+^ was not included in most simulations. In the simulation with two flat bilayers that included ca^2+^, we frequently observed membrane binding through the ca^2+^binding loops of the C_2_ domains (e.g. Figure 1F), and we also observed such interactions in the simulation of a vesicle and a flat bilayer with ca^2+^-bound C_2_AB dissociated from the SNAREs, which we removed in the revised manuscript (see answer to point 1b above). In the simulation of a vesicle and a flat bilayer with C_2_AB bound to ca^2+^ and to the SNAREs, we positioned the C_2_A domain so that its ca^2+^-binding loops can readily interact with the flat bilayer (Figure 5A), and such interactions were often observed during the simulation (e.g. Figure 5B). One of the ca^2+^-binding loops of each C_2_B domain was inserted into the flat bilayer soon after the start of the simulation and remained inserted (Figure 5D, F, H, J), but the other loop could not insert into the membrane because the C_2_B domain did not dissociate from the SNAREs, remaining approximately parallel to the flat bilayer.

b. How does one reconcile that the aliphatic loops on Synaptotagmin C2B domain do not insert into the membrane upon calcium binding as observed in previous structural/functional studies?

See answer to point 4a above.

Reviewer #1 (Recommendations for the authors):1) It is difficult to assess if the results from simulations are a true representative of the biological process or an outcome of the initial condition/constraints chosen. For example, it is puzzling that there is no intra-molecular assembly of the SNAREpins during the simulation even though the coil-coil interactions are expected to occur in the simulation time scales. It appears, in almost all cases, that the SNARE zippering is unaltered at the end of the simulation. Also, it might be possible that the authors' choice to model the juxtamembrane region as a fully unstructured region prevents membrane fusion under the current simulation conditions. While it is not clear if the SNAREs zippering extends through the juxtamembrane (JM) region into the transmembrane region as observed in the crystal structure of full-length SNARE (Stein et al. Nature 2009), it stands to reason the JM region needed to be at least partially structured for effective force transfer to catalyze merging of the bilayers.

We would like to ask the reviewer to keep an open mind about the various issues raised here:

Yes, one might hypothesize that zippering of the C-terminus of the four-helix bundle would be fast once the N-terminal half is assembled, but this prediction is not supported by the simulations. In retrospect, this is not so surprising when one looks at these systems and realizes their complexity, with many interactions that can impose energy barriers to conformational transitions necessary for full zippering (we invite the reviewer to look at some of the snapshots that are now available on Dryad at https://datadryad.org/stash/share/BvqW82678udTxtZ3ifYsx9H2LXy_ttCwm271QmVXvoI). In our opinion, these results are teaching us features that we did not consider previously because we did not have a three-dimensional model of a plausible configuration of this system to look at. This is one of the reasons why we believe that the simulations will be of interest to researchers in the field, even if we did not observe the initiation of fusion in our simulations.

We want to clarify that we did not choose to model the juxtamembrane (JM) regions as fully unstructured sequences. We simply did not impose constraints to keep a helical structure in the JM regions. The notions that the energy of formation of SNARE complex needs to be transferred to the membranes to induce membrane fusion and that the JMs form continuous helices with the SNARE motifs and TM regions to exert this force transfer are hypotheses that have not been demonstrated even though they are widely accepted in the literature. In fact, there are very strong arguments against models invoking continuous helices in the JM regions, as we now discuss:

(lines 442-457): ‘However, assembly of the four-helix bundle brings the membranes within a few nm from each other, and it is unclear how the SNAREs exert additional force on the membranes to induce fusion. A major problem with the widespread notion that the SNARE motif, juxtamembrane linker and TM region of synaptobrevin and syntaxin-1 form continuous helices that force fusion as they zipper from the N- to the C-terminus (Hanson et al., 1997a; Sutton et al., 1998; Weber et al., 1998) is that the bent conformations of the linkers envisioned in these models are unrealistic from an energetic point of view. Although optical tweezer data suggested that interactions between the juxtamembrane linkers contribute to exerting force on the membranes to induce fusion (Gao et al., 2012) helix continuity in the linkers is not required for neurotransmitter release (Kesavan et al., 2007; Zhou et al., 2013). Substantial release was observed even upon insertion of a five-residue sequence into the synaptobrevin linker (Kesavan et al., 2007) despite the fact that this sequence should break the register of linker-linker interactions and contained two (helix disrupting) glycine residues. Note also that the optical tweezer data were obtained in the absence of membranes and that, in vivo, the linkers are likely to interact with the lipids given the proximity of each linker to the adjacent membrane and the abundance of basic residues in the linker sequence (and aromatic residues in the case of synaptobrevin). The extensive interactions of the linkers with the membranes observed in all our simulations support this prediction.’

We want to emphasize that we do not really conclude that the linkers are unstructured and we now make it clear that there are alternative possibilities, but unstructured linkers provide a natural, unbiased starting point for the simulations. These issues are presented in the following sentences:

(lines 458-465): ‘Based on conformational grounds, it is not surprising that the juxtamembrane linkers became unstructured during the simulation that we performed to generate a trans-SNARE complex starting from the crystal structure of the cis-SNARE complex. Note however that we did not perform a systematic analysis to examine the range of linker structures that are compatible with the geometry of a trans-SNARE complex. In any case, the linkers are expected to be unstructured before SNARE complex assembly, which is supported by EPR data (Kim et al., 2002). Therefore, configurations with unstructured linkers were natural, unbiased starting points for the simulations, and helical conformation could be adopted by the linkers during the simulations if they were preferred.’

It is also worth noting that coarse-grained simulations using the MARTINI force field appeared to support the notion that continuous helical conformations of the SNAREs are critical to induce membrane fusion (Risselada et al., 2011), but this result could be a major potential artifact of the approach used. Thus, in the MARTINI force field the secondary structure of proteins is constrained to be mostly fixed due to the lack of explicit hydrogen bonding. Therefore, any reaction that involves a partial (un)folding mechanism would be missed, and the fusion observed in the simulations may have been caused by the helical restraints intrinsic to the force field. We now explain better these issues in the revised manuscript:

(lines 168-181): These models were supported by coarse-grained MD simulations that used the MARTINI force field and modeled the SNAREs in continuous helical conformations (Risselada et al., 2011). However, the intrinsic helical restraints enforced by the force field might bias the results and/or obscure the potential role of conformational changes in the dynamical coupling of the SNAREs to membrane fusion. Moreover, the bending of the helices required to form trans-SNARE complexes leads to unrealistic conformations that are expected to be unfavorable energetically because of their distorted geometry and are not commonly observed in protein structures. Thus, the helical restraints might have played a key role in membrane fusion in these simulations. Although continuous helices were observed in the crystal structure of a cis-SNARE complex that represents the configuration occurring after membrane fusion (Stein et al., 2009), the natural expectation is that the helical structure must break somewhere to accommodate the geometry of a trans-SNARE complex, most likely at the juxtamembrane linker. This expectation has been supported experimentally (Kim et al., 2002) and with all-atom MD simulations (Bykhovskaia, 2021). Moreover, helix continuity in the linkers is not required for neurotransmitter release (Kesavan et al., 2007; Zhou et al., 2013).

Finally, it is also worth noting that there are alternative models that do not envisage a force transfer by the SNAREs on the membranes but rather propose that the SNAREs bring the membranes together and then membrane fusion is catalyzed by bilayer perturbations caused by insertion of hydrophobic sequences, for instance from Sec17 in yeast vacuolar fusion or from the synaptotagmin-1 C_2_ domains in neurotransmitter release [Wickner and Rizo (2017) Mol Biol Cell 28, 707]. Indeed, a recent paper showed that liposome fusion can be observed upon crippling three SNAREs such that C-terminal zippering of the four-helix bundle is impossible when the reactions included Sec17 and Sec18 (the yeast homologues of SNAPs and NSF), and fusion required a hydrophobic loop of Sec17 that inserts into membranes. While it is unclear to what extent these results are relevant to ca^2+^-triggered neurotransmitter release, they do support this alternative model, and it is also known that ca^2+^-dependent insertion of the ca^2+^-binding loops into membrane is critical to trigger release [e.g. Fernandez-Chacon et al. (2001) Nature 410, 41; Rhee et al. (2005) PNAS 102, 18664].

In summary, there are strong reasons to keep an open mind about the mechanism underlying SNARE-dependent membrane fusion.

2) A major conclusion of the report is that the steric clash between Complexin accessory helix and vesicle serves as the fusion clamp and indeed drives the positioning of the SNARE and Synaptotagmin on the planar bilayer. However, there are a couple of factors that might alleviate or even mitigate this steric clash: (i) the vesicle and bilayer are positioned at ~2.3 nm apart at the beginning of the simulation. However, high-resolution cyroEM analysis in synaptosomes/cultured neurons (Fernandez-Busnadiego, R. J Cell Biol 2013; Radhakrishnan et al. PNAS 2021) show that the inter-bilayer distance of docked/primed vesicle is ~4.5 nm. Thus, it might be imperative to carry out the simulation with the physiological accurate inter-bilayer distance.

The initial distance between the vesicle and the bilayer that we used (~2.3 nm) was based on extensive EM studies of synapses performed with high-pressure freezing that observed many vesicles close to being in contact with the plasma membrane and assumed that docked/primed vesicles were within 0-5 nm from the plasma membrane [Imig et al. (2014) Neuron 84, 416]. The Radhakrishnan et al. PNAS 2021 paper that described a detailed analysis of vesicle-plasma membrane distances in synapses analyzed by cryoelectron tomography was published during the course of this work. The data in this paper show that there is a distribution of distances and vesicles are almost in contact with the plasma membrane in some of the images of the paper (e.g. top left of Figure 2A and top right of Figure 3A in that paper). The 3D map constructed from a subset of subtomograms yielded a distance of 3.5 nm between the vesicle and the plasma membrane. Considering the limited resolution of the map (4.4 nm) and the overall distribution of distances observed, these data are not inconsistent with the Imig et al. data. Overall, it seems clear that there is a considerable uncertainty in the distance between a primed vesicle and the plasma membrane.

We agree that it would be desirable to perform additional simulations with different distances between the vesicle and the flat bilayer. However, without including a protein that can bridge the membranes and keep them apart, such as Munc13-1, it is most likely that the SNAREs would still bring the membranes into contact. We do plan to investigate this issue in the future but, as explained above, it will take a long time to apply for and obtain computing time on Frontera to perform these additional studies. To address the concern expressed by the reviewer, we now discuss the uncertainty in the vesicle-plasma membrane distance:

(lines 541-556): ‘While all these observations support the proposal that the consistent overall configuration of the Syt1 C_2_B domain-SNARE-Cpx1(27-72)-flat bilayer assembly observed for the eight primed complexes in the two simulations resembles that present in primed vesicles, there are clear uncertainties with regard to the extent of C-terminal SNARE zippering and the vesicle-flat membrane distance, which are closely related. In our two simulations, the vesicle was drawn into contact with the flat bilayer. However, other factors in addition to complexin are likely to hinder zippering in vivo, most notably Munc13-1 because in the absence of ca^2+^ it bridges membranes in approximately perpendicular orientations that keep the membranes apart (Camacho et al., 2021; Grushin et al., 2022; Quade et al., 2019). Analyses by high-pressure freezing electron tomography (ET) (Imig et al., 2014) and cryo-ET (Radhakrishnan et al., 2021) showed that docked vesicles exhibit a distribution of distances from the plasma membrane that range from 0 to several nm, and a density map built from a subset of subtomograms revealed a distance of 3.5 nm. However, a recent model that can explain a large amount of available presynaptic plasticity data invoked two primed states, one that involves partially assembled SNARE complexes and has low release probability (loose state), and another with more fully assembled SNARE complexes that has a much higher release probability (tight state) (Neher and Brose, 2018). Hence, it is plausible that the vesicles that are closest to the plasma membrane account for much of the neurotransmitter release observed.’

Please note also that, as explained in our response to point 3b of essential revisions, the overall configuration of the Syt1 C2B domain-SNARE-Cpx1(27-72)-flat bilayer assembly is compatible with vesicle-plasma membrane distances ranges from 0 to a few nm, which is now illustrated in Figure 6D.

(ii) Complexin molecule has been positioned on SNAREs assuming a fully-zippered SNARE complex. However, there is sufficient evidence that SNAREs are likely only partially-assembled in an RPP vesicle (Hua & Charlton, Nat Neurosci 1999, Prashad & Charlton, PLoS One, 2014), and the positioning of the CPX, esp. the accessory helix is correlated to the extent of SNARE assembly (Choi et al. ELife 2016, Kummel et al. Nat Struct Biol 2011; Zhou et al. Nature 2017). Furthermore, accessory helix has been shown to interact with c-terminal ends of t- and v-SNARE molecules (Kummel et al. Nat Struct Mol Biol 2011; Malsam et al., Cell Reports 2020). Thus, it is possible that the alternate positioning of the accessory helix and other interactions might reduce the observed steric clash.

As explained above, the analysis from Neher and Brose 2018 indicates that a large amount of physiological data can be explained by a model with two states that have different extents of C-terminal zippering, and most ca^2+^-triggered release occurs from the state that is more zippered. We do not know the extent of zippering in the tight state, but it is plausible that zippering is similar to that observed in our simulations. We agree that the position of the complexin-1 accessory helix may depend on the extent of C-terminal SNARE complex assembly, but please note that we observed fast motions of the accessory helix during the simulations, which is particularly well illustrated by the changes observed during 336 ns in Figure 3—figure supplement 2. Hence, there was ample freedom for the accessory helix to sample alternative positions and the behavior that we observed is consistent with the data suggesting alternative positions of the accessory helix presented in the papers cited by the reviewer. It is also important to note that, even if the SNARE complex is less zippered in the primed state than proposed in our model, as is the case for some of the complexes in our simulations, there still can be steric clashes of the helix with the vesicle. Even if there are less steric clashes, there would be more clashes as soon as the SNARE complex zippers further to induce fusion, so steric clashes would still hinder release. We now discuss this issue in the following sentences:

(lines 573-580): ‘Manually moving the vesicle in the pose of Figure 6B to a distance of 3 nm from the flat bilayer (Figure 6D) shows that the same spring-loaded configuration of the Syt1 C_2_B domain-SNARE-Cpx1(27-72)-flat bilayer assembly could be kept by stretching the synaptobrevin juxtamembrane linker, and the Cpx1(27-72) helix would still hinder progress toward final zippering and fusion. This configuration is also largely compatible with longer vesicle-flat bilayer distances if the linker is stretched further and/or there is partial SNARE unzippering. In this case, steric clashes of the Cpx1(27-72) helix with the vesicle might be alleviated or eliminated, but they would occur as soon there is full zippering and hence would still hinder vesicle fusion.’

The Malsam et al., Cell Reports 2020 did provide some evidence of interactions of the accessory helix with SNAP-25, but note that replacement of the accessory helix with an unrelated helical sequence still retained its inhibitory activity fully [Radoff et al. (2014) e*Life* 3, e04553] (see answer to point 3c of Essential Revisions above). This observation argues against an important role for specific complexin-SNARE interactions in inhibiting release. In any case, non-specific interactions could play a role, and our simulations did not preclude interactions of the complexin-1 accessory helix. In fact we did observe such interactions occasionally, as we now show in a new figure panel (Figure 3—figure supplement 2G). We now address these issues in the discussion:

(lines 531-540): ‘Note also that we observed occasional interactions of the Cpx1(27-72) accessory helix with C-terminal residues of the synaptobrevin and SNAP-25 SNARE motifs that are favored by proximity (e.g. Figure 3G, Figure 3—figure supplement 2G), and recent cross-linking experiments suggested that very weak interactions of the complexin accessory helix with synaptobrevin and SNAP-25 hinder C-terminal zippering and release (Malsam et al., 2020). It is unlikely that specific interactions underlie the complexin inhibitory function, as this function was retained when the accessory helix was replaced with an unrelated helical sequence, and helix propensity appears to be the key determinant for the inhibitory function (Radoff et al., 2014). However, very weak complexin-SNARE interactions that do not need to be specific may slow down C-terminal SNARE zippering and thus contribute also to inhibition of release by the complex accessory helix.’

3) How does one reconcile that the aliphatic loops on Synaptotagmin C2B domain do not insert into the membrane upon calcium binding as observed in previous structural/functional studies (Grushin et al. Nat Comms 2019; Kuo et al. J Mol Biol 2009) even though synaptotagmin interacts with the membrane, including partial insertion of the C2B aliphatic loop, under calcium-free conditions. This is a rather crucial and missing piece considering that calcium-triggered membrane insertion is predicted to be the driving force for triggered fusion.

Please see also answer to point 4a of Essential revisions above.

Reviewer #2 (Recommendations for the authors):1. The authors' major conclusion is that the AA simulations support the model of Voleti et al. for the organization that clamps fusion in the pre-ca primed state. However, from Figures 3, 4 (and associated figure supplements) fusion seems very likely not to be clamped, given the vesicle contacts the planar membrane (the degree of contact is still growing at the end of the simulation, Figure 4 supp 4). As stated in (lines 420-425) the vesicle membrane is not flattened. This indicates a lower force than with SNAREs alone, but seems unlikely to block fusion. Due to running time limitations, AA simulations cannot test if fusion would occur in a physiological time. The structure does not keep the membranes apart, as it rotates and permits contact. The authors are clear about this – indeed, to predict the orientation is stated as a major objective. But the conclusions of lines 365-367 and the final sentence of the abstract, suggesting these results demonstrate a fusion clamp, seems unjustified as far as I can see. The emphasis on the cpx accessory helix role also appears somewhat exaggerated, as if on its own it provides a mini-buttress that separates vesicle and planar membrane. It's hard for me to imagine it supports much force in this configuration.

Please see the responses to point 2b, 3b and 3c of Essential revisions, as well as to point 2 from reviewer 1.

2. The simulations with bound calcium (final section of Results) seem inconclusive. The number of contacts is still growing at the end of the simulation, and we cannot know if the C2B will ever dissociate from the SNAREs. It's very reasonable to try this simulation but given the outcome I'm not sure a long section is merited, particularly with the tentative title "Potential effects of Ca2plus binding to synaptotagmin-1." This negative, albeit interesting finding, might be briefly summarized in the main text.

We agree that the simulations with ca^2+^ did not yield very conclusive data and have removed the last simulation in which the synaptotagmin-1 C_2_AB fragment was dissociated from the SNAREs, shortening this section considerably. We prefer to keep the simulation started with C_2_AB bound to ca^2+^ and to the SNARE complex because it illustrates two points: i) that no ca^2+^-induced dissociation of C_2_B from the SNAREs occurred in the time scale of this simulation for any of the four complexes; and ii) that there was lower tendency of the SNARE complex to interact with the flat bilayer because of the absence of clashes between complexin-1 and the vesicle. We have kept this description as a small section (with a different title) because the previous section is very long. We hope that this is acceptable and would be willing to remove this section if the reviewer feels strongly about it. As we do throughout the paper, we do not draw firm conclusions from these results but point out what the data suggest.

3. The manuscript would be strengthened by a more balanced presentation acknowledging the limitations of AA simulations (while of course still extolling their merits) and connecting to some degree with analysis on other scales, including coarse-grained approaches beyond MARTINI. SNARE-mediated fusion was studied using ultra coarse-grained (Mostafavi et al., 2017; McDargh et al., 2018) and even continuum (Manca et al., 2019) representations. Every approach has strengths and weaknesses. AA approaches scrutinize local issues as no others can, but presently they are remote from being able to demonstrate hemifusion, fusion, unclamping and ca-evoked fusion. Making matters worse, NT release is clearly stochastic, so multiple runs are needed for each condition. These limitations are apparent in this study: almost every conclusion comes with a caveat related to running time. In previous seminal MARTINI studies that achieved fusion (Risselada, Sharma and Lindau) the conditions were intentionally biased for fusion (vesicle size, lipid composition, temperature, helical LDs) or nanodiscs were used. In (Risselada, 2011) no fusion was observed when the LD was made unstructured.

We fully agree with this criticism and now present a more balanced perspective. Please see our responses to point 1b of Essential Revisions and the introductory comments from reviewer 2 above.

In the Introduction the authors assert that experiments suggest "..the fusion step occurs in just a few microseconds," which timescales AA simulations may be able to access. They quote the 60 microsec delay times (ca influx to first sign of the excitatory post synaptic current, EPSC) reported by Sabatini and Regehr at 38 degrees. However, 0.5 -2 ms is much more typical in the literature (admittedly, the 38 degrees study is distinguished by the temperature being physiological.) Related, long-ago Katz argued other processes (e.g. NT diffusion across the synaptic cleft) are much faster than NT release (Katz and Miledi, 1965).

Please see our response to point 2b of Essential revisions above.

4. In simulations with SNAREs only, the SNARE complexes are cleared laterally, and the membranes are squashed together, generating an ECZ (extended contact zone, a flat portion of vesicle), Figures 1C, 1F. This is precisely the behavior seen in highly coarse-grained simulations (Mostafavi et al., 2017, Mcdargh et al., 2018), where entropic SNARE-SNARE and SNARE-membrane forces cleared the fusion site and pressed the vesicles together (those studies used undeformable membrane surfaces, so no vesicle flattening occurred). The entropic forces were predicted to provoke fusion after a time of order msec, with faster fusion for more SNAREs. These coarse-grained simulations and their relation to the present findings should be discussed.

We now mention that extended contact interfaces can arise from the entropic forces mentioned by the reviewer in the discussion:

(lines 469-471): ‘Coarse-grained simulations have suggested that formation of such interfaces can also arise from entropic forces that favor outward movement of the SNARE complexes, away from the center of the interface (Mostafavi et al., 2017).’

With regard to the issue of the time scale of fusion of these interfaces, please see response to point 2b of Essential revisions.

The authors suggest the pressing together of the membranes is caused by binding of the LDs to the vesicle membrane (lines 262-266). This does not seem a plausible alternative to the proposed entropic forces, as LD-membrane adhesion would not favor the SNAREs being pushed outwards as far as I can see.

We believe that the linker-membrane interactions may contribute to bring the membranes together but we agree that they would not favor pushing the SNAREs outwards. We have softened the statement regarding the potential role of these interactions in bringing membranes together in the Results section:

(lines 204-206): ‘these findings suggest that any electrostatic repulsion existing between the SNARE four-helix bundle and the membranes can be readily overcome by the high stability of the SNARE four-helix bundle and perhaps some contribution from the linkerbilayer interactions’.

The authors argue that the ECZ in the SNARE-only simulations suggests SNAREs alone cannot fuse membranes rapidly, since fusion was slow in Hernandez et al., 2012 and Witkowska et al., 2021 where ECZs were seen. However, in those in vitro studies many other processes preceded fusion (SNARE assembly, docking etc) and micron scale GUVs were used by Witkowska et al.

As we explain in the response to point 3b of Essential Revisions, the contact interfaces observed by cryo-EM in Hernandez et al. 2012 occurred early in the liposome fusion reactions and gradually disappeared over the time course of the reactions (minute time scale). Hence, at least a fraction of liposomes that had extended contacts took minutes to fuse. This conclusion is consistent not only with the GUV data from Witkowska et al. 2021 but also with fluorescence correlation spectroscopy data showing that liposome docking preceded fusion and that a majority of liposomes fused minutes after docking [Cypionka et al. (2009) PNAS 106, 18575], which we now mention in the paper.

5. A concern is the presentation, whose clarity would benefit from a more concise text. It is laudable to convey the details (computational papers where readers cannot tell what was done are frustrating), but many passages are long repeats of previous passages. For example, opening paragraphs of sections in Results often repeat descriptions of simulations in previous sections at great length, then specifying what was different in the current section. These could be massively shortcut. Short summaries in the main text, with details left to Methods or Supplementary materials, would be more digestible for readers.

We have shortened the Results section by five pages, merging the descriptions that overlapped with the methods section into this section.

6. This paper describes many current hot issues in the field, a great service. The figures are very nice but would be helped by a simple visual key to identify β sandwiches, the polybasic face, ca-binding loops, etc. For an uninitiated reader, it is tough staring at these protein structures trying to figure out which features are where? Also, I suggest adding a length bar to one or more Figures

We have redesigned Figure 1—figure supplement 1 to increase the size of the ribbon diagram of the synaptotagmin-1 C_2_AB fragment that was in panel E and now is in panel A so that the reader knows where its key elements are from the beginning. In the diagram we have included stick models of the side chains that form the polybasic face and the primary interface, and we have labeled the side chains so that the reader has a guide for their location. The SNARE complex provides a good reference for sizes, as its length is 11 nm. We have put a 10 nm length bar in Figure 1A for the benefit of readers who are less familiar with the SNARE complex.

7. It is stated that the Cpx accessory helix inhibits release "likely" because it causes steric clashes with the vesicle (line 85). I think this is a powerful and very reasonable suggestion, but perhaps "possibly" would better reflect current uncertainty about the mechanism.

We have replaced ‘likely’ with ‘possibly’ (line 79 now).

8. Their 26 nm diameter vesicles are ~ 2-fold smaller than synaptic vesicles. I do understand why this measure is taken (and the authors mention why), but the synaptic vesicle size should be stated.

The vesicle size is stated in the same paragraph describing the 26 nm vesicle that we built (lines 222-233).

Reviewer #3 (Recommendations for the authors):My overall sense from this study is that the simulation efforts are preliminary and sufficiently incomplete to cause concern about the validity of the conclusions. I am concerned about several omissions and their potential impact on conclusions about the prefusion complex and the possible trajectories leading to fusion:1. SNARE/Syt/Cpx omissions – What is the potential impact of removing the Habc region of syntaxin 1 given its significant excluded volume and potential interactions with membrane PIP2? Similarly, excluding the palmitoylated linker regions of SNAP-25 may play important and interesting roles affecting SNARE orientation, the distribution of forces between SNAREs and membranes, and membrane behavior. The lack of a Syt1 juxtamembrane region (as well as its transmembrane anchor) seems like a real missed opportunity given past work suggesting several interesting hypotheses for intramolecular and membrane interactions of this region. Finally, omitting the C-terminal domain of Cpx1 with its known membrane-interacting region may have significant implications for the detailed behavior of Cpx1 and the forces acting on its SNARE-binding region. While no realistic simulation could currently hope to capture all of this, I would have preferred fewer simulations with more assessment of whether or not some of these omissions would cause major changes to the behavior of the simulated system.

We agree that some components that we omitted may affect the mechanism of membrane fusion and we now emphasize this in the manuscript, as explained above. However, we note that several of the regions mentioned by the reviewer (SNAP-25 linker, Syt1 juxtamembrane, complexin-1 N- and C-terminal regions) are expected to be largely unstructured even if they form small elements of secondary structure, and there is very limited information on how they are arranged with respect to the rest of the corresponding protein, if there is any specific arrangement. Modeling these regions in arbitrary conformations could yield energy barriers that would further compound the problem with time scale that is a key challenge in these simulations. We also note that the syntaxin-1 H_abc_ domain is crucial for synaptic vesicle priming because it is central for the Munc18-1-Munc13-1-dependent pathway that leads to SNARE complex assembly [Ma et al. (2013) Science 339, 421], but the release probability is not substantially affected by deletion of the H_abc_ domain [Vardar et (2021) *ELife* 10, e69498], suggesting that this domain is not involved in the ca^2+^-triggered fusion step or in the primed complexes that are ready for fast release.

As stated above, we hope the reviewer realize that many important contributions in our field were made with experiments using minimal systems that included even fewer components than our simulations.

2. Calcium-phospholipid interactions – When the authors included 5 calcium ions per Syt1 to assess the impact of elevated local calcium on the simulation dynamics, I was struck by a lack of corresponding calcium interactions with PS and PIP2. 20 calcium ions in the simulated volume would roughly correspond to 1 mM calcium, and even that wouldn't necessarily lead to all 20 potential binding sites on Syt1 being occupied. At the same time, one would expect divalent interactions with PS and PIP2, which could neutralize membrane repulsion and significantly lower at least one aspect of the complex membrane fusion energy barrier. Work by chemists such as Feigenson have indicated strong calcium-mediated interactions between even PS and PC at concentrations much lower than 1 mM (Biochemistry 1989). Some of these chemical details may not be capable of proper simulation in the MD formalism deployed in the current study, but this should be addressable in some fashion.

Much of the work was performed without ca^2+^ and all main results that we stress were obtained without ca^2+^. Please see also the response to point 4a of Essential Revisions.

3. I was not convinced by the authors' reasoning regarding one microsecond being a relevant timescale for synaptic vesicle fusion. And given that even some initial phase of membrane fusion was not observed in these simulations, I find it impossible to access wherein the process of priming/fusion these current simulations reside. The fastest reported latency between presynaptic calcium entry and fusion is around 60 microseconds as the authors point with the Sabatini/Regehr study. Importantly, that was not a single-synapse measurement but instead, a population measure involving 1000s of synapses. So the first latency likely represents a small population from the fast tail of a distribution of fusion times. And given the 1-2 microsecond delay for cleft glutamate diffusion and the 10-20 microsecond activation time of a stellate cell AMPA receptor, it is likely that the calcium-fusion delay at this synapse resides in the 50-100 microsecond time window. Thus, a 400-nanosecond simulation would seem far too brief to do this process justice.

Please see the response to point 2b of Essential revisions.

4. Since the simulations are certainly not trying to capture relevant roles and impacts of other core synaptic proteins such as Munc13 and Munc18, I thought that the use of 'primed' state was a bit oversold and misleading in this manuscript. These simulations seem most appropriate for interpreting in vitro liposome fusion experiments utilizing just SNAREs or SNAREs plus Cpx/Syt1. I am not sure what it would mean to describe a primed state for the SNAREs and synaptic vesicle without also having Munc13 present and bound at least to the two membranes if not also to the SNAREs. I appreciate that the authors are modeling something that represents our best guess for the SNARE assembly on a tightly docked and primed vesicle, but this simulation clearly lacks crucial elements that go into what the field usually refers to as a primed synaptic vesicle. I would want the language used to reflect this as much as possible.

Please see the response to point 3a of Essential revisions.

5. I do not have a sense for how worrisome it is from a technical perspective to forgo replicate simulations. For instance, is it better to have two replicates each of three simulations rather than six slightly different simulations each done once? It would be useful to have some discussion of the uncertainty/reliability attached to these conclusions given the absence of replicates.

Please see General comments above.

[Editors' note: further revisions were suggested prior to acceptance, as described below.]

The manuscript has been improved but there are some remaining issues that need to be addressed, as outlined below.Please address the points brought up by Reviewer #3 on the primed state and on calcium/lipid interactions, at least at the writing level. To avoid ambiguity, it would be better to remove the word 'primed' from the title, lines 34 and 37 in the abstract, and line 129 in the introduction, and change the corresponding sentences when needed. This is probably not sufficient because there are so many mentions of 'primed states', primed complexes', or 'vesicle priming'.

Please see response to the first point from reviewer #3.

Regarding the interactions of calcium with lipids, we realize that getting computational time is a limiting and costly resource currently. Asking to perform additional simulations involving lipid/calcium interactions may be difficult but the authors can certainly mention that it is a limitation of their simulations that may affect the outcome and should be tested in the future.

We mentioned and still mention the limitations arising from the limited time of our simulations in the abstract, the introduction and the discussion. See also the response to the second point from reviewer #3.

Reviewer #3 (Recommendations for the authors):Rizo and colleagues have shortened and edited their manuscript as requested in the first review. I wasn't strongly enthusiastic about this MD study during the first round and remain somewhat dissatisfied after reading the authors' response to our concerns. Two of my concerns were largely ignored by the authors in their rebuttal but remain worrisome to me nonetheless.One concern is the authors' continued declaration that their simulations are synonymous with the primed fusion complex. Vesicle priming is already a somewhat muddled concept in the field and this manuscript doesn't help the confusion. I appreciate that they edited their title a bit but anyone glancing at the paper or searching for it on PubMed would very likely interpret this as the primed state prior to fusion. In addition, they conclude in the abstract (line 37) that 'the primed state contains macromolecular assemblies …' whereas I don't believe the simulations warrant this conclusion. This is reiterated in the last sentence of the introduction (line 129-132) but at least they soften the conclusion with 'suggest that'. I am not sure the authors got that much more out of the model than they put in to begin with since they chose starting points that they were already convinced represented their best guess at the primed state of the fusion complex. Perhaps some of the observations regarding the juxtamembrane linkers of the SNAREs are moderately unexpected, but given that no fusion was witnessed, the reader doesn't know which details of the current model truly correspond to relevant prefusion scenarios.

Both in the paper and the rebuttal letter, we provided multiple arguments supporting the notion that the simulations with synaptotagmin-1-SNARE-complexin-1 complexes are relevant to the primed state of synaptic vesicles. The reviewer is reluctant to accept this relevance (and even the concept of a primed state), but does not provide specific reasons against any of our arguments. It is not only us that are convinced that the primed state of synaptic vesicles includes trans-SNARE complexes bound to synaptotagmin-1 and complexin-1 through the same modes as the initial state that we built; this view is based on extensive evidence and is shared by many people in the field, including leaders such as James Rothman, Axel Brunger, Erwin Neher and Nils Brose among others [e.g. Grushin et al. (2019) Nat Commun 10, 2413; Radhakrishnan et al. (2021) PNAS 118, e2024029118; Brunger et al. (2018) Annu Rev Biophys 47, 469; Neher and Brose (2018) Neuron 100, 1283]. Note also that the reviewer trivializes our results, as the main purpose of the simulations with synaptotagmin-1-SNARE-complexin-1 complexes was to investigate how they are oriented with respect to the membranes, and our simulations yielded consistent results for eight out of eight complexes in this respect.

We are aware of the limitations of our study and we emphasize them prominently in the abstract, introduction and discussion, normally using verbs likes suggest rather than demonstrate. In line 37 of the abstract we did not use ‘conclude’, as indicated by the reviewer; the corresponding sentence started in line 35 and used the expression ‘suggest that’. We need to somehow relate the simulations to the existing knowledge in the field; otherwise, readers will be completely confused, will not know why we performed these simulations and will not understand the potential implications of our results. To partially address the concern from the reviewer, we have checked the manuscript carefully to make sure that we do not use strong words like demonstrate or show when referring to the relevance of the results to vesicle priming, and we have changed the title to:

‘All-atom molecular dynamics simulations of Synaptotagmin-SNARE-complexin complexes bridging a vesicle and a flat lipid bilayer’

We hope that this is acceptable.

My other concern is that the authors continue to ignore the very real possibility that calcium interactions directly with the phospholipids (independent of Syt1 C2 domains) are a critical aspect of membrane fusion. This has been studied chemically and using in vitro membrane fusion assays for 50 years but wasn't even discussed as a possible explanation for the lack of fusion in the simulation where calcium was included. Just to be explicit, I am thinking of papers such as Papahadjopoulos BBA 1976, Feigenson Biochem 1986,1987, and 1989 studies, Kachar Biophys J 1986, and modern studies such as Churchward Biophys J 2008. While I don't know what the technical limitations of implementing calcium-phospholipid interactions are in all-atom MD, I can find examples in the literature such as Allolio and Harries ACS Nano 2021 and Allolio et al. PNAS 2018 where calcium ion interactions with phospholipids during membrane fusion are explicitly incorporated, so I assume there isn't a fundamental reason this cannot be explored or acknowledged. I don't think it would be surprising if some of the key results here such as the juxtamembrane linker electrostatic interactions with the membrane would be strongly affected in addition to the possibility that the fusion energy barrier would be lowered sufficiently to witness the beginning of a fusion event on a microsecond time scale.

We apologize because in the previous round of review we thought that the reviewer was referring to interactions between lipids and ca^2+^ ions bound to the synaptotagmin-1 C_2_ domains. Such interactions are known to dramatically increase the affinity of synaptotagmin-1 for ca^2+^ and hence are intrinsic part of the role of synaptotagmin-1 as a ca^2+^ sensor for release.

We did not realize that the reviewer was referring to interactions of lipids with free ca^2+^. Some scientists indeed considered the possibility that such interactions mediate synaptic vesicle fusion many years ago. However, measurements at the calyx of Held showed that 10 µM ca^2+^ triggered release of 80% of available vesicles [Schneggenburger and Neher (2000) Nature 406, 889]. Moreover, it was established that synaptotagmin-1 acts as the major ca^2+^ sensor that triggers release and the apparent ca^2+^ affinity of synaptotagmin-1 in phospholipid binding correlated well with the ca^2+^-dependence of release [FernandezChacon et al. (2001) Nature 410, 41]. These findings argue against a role for interactions of phospholipids with free ca^2+^ in release, as the affinity of these interactions is much weaker than that of synaptotagmin-1 in the presence of phospholipids and does not explain the ca^2+^ dependence of neurotransmitter release.

Lamellar phases of phosphatididyl serine (PS) can bind to ca^2+^ at low µM concentrations due to cooperativity effects, but membranes with a lipid composition more similar to synaptic membranes bind ca^2+^ with much lower affinity. For instance, 5 mM ca^2+^ was used to observe ca^2+^-induced clustering of liposomes containing 25% PS by electron microscopy in the Kachar et al. (1986) reference mentioned by the reviewer (Biophys. J. 50, 779), but no clustering of liposomes containing 30% PS was observed by dynamic light scattering in the presence of 1 mM ca^2+^ [Arac et al. (2006) Nat. Struct. Mol. Biol 13, 209]. The reviewer mentions papers describing simulations that incorporated ca^2+^, and these papers indeed reported ca^2+^-induced membrane fusion. However, Allolio et al. (2018) [Proc Natl Acad Sci U S A 115, 11923] included 450 mM ca^2+^ in the simulations, and Allolio et al. (2021) [ACS Nano, 15, 12880] used ca^2+^ to lipid ratios of 1:10 or higher, which in our qscv system with SNAREs alone would translate to 14.4 mM ca^2+^ or higher. We believe it is highly questionable that results obtained with such high ca^2+^ concentrations are physiologically relevant. We also note that, if we wanted to use the 10 µM ca^2+^ concentration that triggers release at the calyx of Held in our qscv system, we would need to add 0.2 ca^2+^ ions in the simulation box. One could claim that binding of ca^2+^ to synaptotagmin-1 could considerably raise the local ca^2+^ concentration, but this ca^2+^ would not be free, and local enrichment of ca^2+^ at the membrane surface is much less favorable because of the much lower ca^2+^ affinity.

We agree with the reviewer that the possibility that binding of free ca^2+^ to membranes might facilitate synaptic vesicle fusion was an interesting idea and we believe that the idea cannot be completely ruled out. However, in our opinion this is not likely and we cannot agree with the reviewer that the absence of free ca^2+^ in our simulations casts serious doubts on our results. We also believe that a substantial discussion of this subject as described above would unnecessarily lengthen the already long discussion. To partially address the comments from the reviewer, we added the following sentence in the final paragraph of the discussion of the revised manuscript (lines 592-595):

‘Furthermore, some evidence suggested many years ago that direct binding of ca^2+^ to phospholipids could trigger synaptic vesicle fusion (Papahadjopoulos et al., 1976) and, although it is now generally believed that ca^2+^ triggers neurotransmitter release by binding to Syt1, it is plausible that ca^2+^-phospholipid interactions might also contribute to trigger membrane fusion.’

Overall, this was a nice first effort at an ambitious simulation scale and could serve as an introductory template for future attempts at modeling SNARE-mediated fusion. The preliminary and underdeveloped feel of the manuscript and notably, the lack of some sort of fusion-like transition captured in the simulations diminish my enthusiasm a bit.

We are glad the reviewer considers our work ‘a nice first effort’.